

# Impacts of physical parametrization on prediction of ethane concentrations for oil and gas emissions

Maryam Abdi-Oskouei[1], Gabriele Pfister[2], Frank Flocke[2], Negin Sobhani[2], Pablo Saide[3], Alan Fried[4], Dirk Richter[4], Petter Weibring[4], and James Walega[4], Gregory Carmichael[1]

[1] Center for Global and Regional Environmental Research (CGRER), University of Iowa, Iowa City, Iowa, USA
[2] National Center for Atmospheric Research (NCAR), Boulder, Colorado, USA
[3] Department of Atmospheric and Oceanic Sciences, University of California Los Angeles (UCLA), Los Angeles, California, USA
[4] Institute of Arctic and Alpine Research, University of Colorado, Boulder, CO, USA

*Correspondence to*: Maryam Abdi-Oskouei (maryam-abdioskouei@uiowa.edu)

**Abstract.** Recent increases in the Natural Gas (NG) production through hydraulic fracturing have called into question the climate benefit of switching from coal-fired to natural gas-fired power plants. Higher than expected levels of methane, Non-Methane Hydrocarbons (NMHC), and NOx have been observed in areas close to oil and NG operation facilities. Large uncertainties in the oil and NG operation emission inventories reduce the confidence level in the impact assessment of such activities on regional air quality and climate, as well as development of effective mitigation policies. In this work, we used ethane as the indicator of oil and NG emissions and explored the sensitivity of ethane to different physical parametrizations and simulation set-ups in the Weather Research and Forecasting with Chemistry (WRF-Chem) model using the U.S. EPA National Emission Inventory (NEI-2011). We evaluated the impact of the following configurations and parameterizations on predicted ethane concentrations: Planetary Boundary Layer (PBL) parametrizations, daily re-initialization of meteorological variables, meteorological initial and boundary conditions, and horizontal resolution. We assessed the uncertainties around oil and NG emissions by using measurements from the FRAPPÉ and DISCOVER-AQ campaigns over the Northern Front Range Metropolitan Area (NFRMA) in summer 2014. The sensitivity analysis shows up to 57.3% variability in normalized mean bias of the near-surface modeled ethane across the simulations, which highlights the important role of model configurations on the model performance and ultimately the assessment of emissions. Comparison between airborne measurements and the sensitivity simulations indicates that the model-measurement bias of ethane ranged from -14.9 ppb to -8.2 ppb (NMB ranged from -80.5% to -44%) in regions close to oil and NG activities. Under-prediction of ethane concentration in all sensitivity runs suggests an actual under-estimation of the oil and NG emissions in the NEI-2011. Increase of oil and NG emissions in the simulations partially improved the model performance in capturing ethane and lumped alkanes (HC3) concentrations but did not impact the model performance in capturing benzene, toluene, and xylene which is due to very low emission rates of these species from oil and NG sector in the NEI-2011.





## 1. Introduction

Recent advances in the unconventional Natural Gas (NG) production technology (hydraulic fracturing) have resulted in economical access to NG reserves in deep shale formations and a 36% rise in US NG production from 2005 to 2014 (Lyon, 2015). Increase in the NG production, decrease in the NG price, and environmental advantages of NG-fired power plants over

coal-fired power plants have made NG an important competitor for coal in the electricity generation sector. In 2015, NG and coal each had a 33% share in the electricity generation in the US. It is predicted that NG's share in electricity generation will grow 1.5% every year (Energy information administration of US Department of Energy., 2016; U.S. Energy Information Administration, 2016). With the rapid increase in the unconventional oil and NG production, higher than expected levels of greenhouse gases, specifically methane, and air pollutants such Non-Methane Hydrocarbons (NMHC) and NOx have been

observed in some places in vicinity of oil and NG facilities. The high concentrations of these chemicals measured in many studies at different scales and regions suggest that official emission inventories (e. g. Greenhouse Gas Inventory (GHGI) and Emission Database for Global Atmospheric Research (EDGAR)) fail to capture the magnitude of emissions from unconventional extraction activities (Brandt et al., 2014). The underestimation of emission inventories has raised concerns regarding the climate implications of promoting NG as the "bridge fuel" (Alvarez et al., 2012; Howarth et al., 2011; Levi,

2013; McJeon et al., 2014), and its impacts on the air quality and public health (Halliday et al., 2016; McKenzie et al., 2012). Additionally, Methane and NMHC emitted from the oil and NG sector can degrade regional air quality and contribute to ozone formation on regional and global scales (Helmig et al., 2016). Outdated Emission Factors (EF), super-emitters in the production systems, and rapid growth in the production facilities are some of the reasons for the underestimation (Brandt et al., 2014; Lyon, 2015; Zavala-Araiza et al., 2015).

The Colorado Northern Front Range (NFR), including the Denver metropolitan area, is located between the Rocky Mountains and the High Plains with a total population of about 4.8 million. In 2007, a large region of the NFR was declared in nonattainment of the National Ambient Air Quality Standard (NAAQS) for 8h average ozone. Major sources of pollutants in this area are vehicle emissions, oil and NG operation, agriculture and feedlots, and power plants. In the past years, oil and NG

development has increased drastically in the NFR. NG production in Weld County has increased from $1.97 \times 10^6$ Thousand Cubic Feet (MCF) to $6.42 \times 10^6$ MCF from 2004 to 2016. The Wattenberg gas field in Weld County is close to populated regions and has the highest well density in the NFR with more than 25,000 active NG wells (Colorado Oil and Gas Conservation Commission, 2017). In the NFR, measured NMHCs are 18-77 times greater than the regional background as determined from the NOAA flask network (Thompson et al., 2014). High levels of NMHC can cause health concerns at

regional scales and can contribute significantly to the ozone pollution in the region (Cheadle et al., 2017; Gilman et al., 2013; McDuffie et al., 2016; Pétron et al., 2012; Thompson et al., 2014). Using box models constrained with observations, McDuffie et al. (2016) estimated that NFR oil and NG activities contribute ~50% to the regional Volatile Organic Compound (VOC) OH reactivity and 20% to the regional photochemical ozone production.



Mass balance approach methods have been widely used to estimate the emissions from oil and NG activities (Karion et al., 2015; Peischl et al., 2016; Pétron et al., 2012; Smith et al., 2015). This method cannot provide details on the spatial and temporal variability of emissions and has limitations in domains with complex atmospheric transport such as the NFR. High

resolution three-dimensional Chemical Transport Models (CTMs) can better capture the variability in meteorology and chemistry in different domains. Paired with observations, CTMs help evaluate the performance of emission inventories on high temporal and spatial scales and allow assessments of the impact of oil and NG activities on regional air quality. Ahmadov et al. (2015) used the Weather Research and Forecasting Model with Chemistry (WRF-Chem) to study high ozone episodes and emission reduction scenarios in the Uintah Basin. Their results show a strong underestimation of methane and VOC

emissions in the National Emission Inventory 2011 (NEI-2011).

WRF-Chem provides users with different dynamical, physical, and chemical schemes (Grell et al., 2005; Skamarock et al., 2008). These choices can impact the performance of the model, specifically in regions with complex transport patterns (Saide et al., 2011). In order to assess the performance of emission inventories, it is critical to address the uncertainties derived from

model configurations on simulated concentration fields. In this work, we quantify the impact of WRF-Chem configurations on predicting the concentration of oil and NG emissions in the NFR. VOCs in the NFR have shown a clear source signature associated with oil and NG activities (Gilman et al., 2013; Pétron et al., 2014). Diverse air pollution sources and complex metrological patterns due to mountain-valley circulation, high elevation, and harsh terrain are some of the challenges for air quality modeling in this area. We use ethane, which has a simple chemical cycle and a lifetime of about two months, as a tracer

for oil and NG (Helmig et al., 2016). We explore the sensitivity of the modeled transport and ethane concentrations to different WRF-Chem physical parametrizations and set-ups. The model and emission inventory performance are evaluated by comparing meteorological parameters as well as ethane and VOC concentrations to surface and airborne measurements. To inform not only about the absolute magnitude in the ethane emissions but to further explore the feasibility to constrain other trace gas oil and NG emissions, we investigate CO and VOC emission estimates from oil and NG sector and VOC ratios in

the observations and the model.

## 2. Method

### 2.1. Aircraft and ground-based observations

The National Science Foundation/National Center for Atmospheric Research (NSF/NCAR) Front Range Air Pollution and Photochemistry Éxperiment (FRAPPÉ) and National Aeronautics and Space Administration (NASA) Deriving Information

on Surface Conditions from COlumn and VERtically Resolved Observations Relevant to Air Quality (DISCOVER-AQ) campaigns were conducted in July and August 2014, in the NFR Colorado. These two campaigns provide detailed and coherent airborne and ground-based measurements in this area, which can assist in evaluation and improvement of chemical transport models and emission inventories. The NSF/NCAR C130 collected extensive airborne measurements of various atmospheric



constituents during the FRAPPÉ campaign. A total of 15 flights (~80 flight hours) were conducted in the NFR with the goal of mapping the emissions and their transport and chemistry in this region. During the DISCOVER-AQ campaign, the NASA P3B aircraft performed approximately 20 flights containing spiral ascents or descents over six key sites in the NFR to capture the vertical profiles of the atmospheric constituents and their diurnal variation.

The National Oceanic and Atmospheric Administration (NOAA), the Colorado Department of Public Health and Environment (CDPHE), and the National Park Services (NPS) operated numerous ground-level measurement sites during these two campaigns. In this work, we present ground-level measurements from the NOAA Boulder Atmospheric Observatory (BAO; 40.05°N, 105.01°W, 1584 m above sea level (asl)), the NOAA Platteville site (PAO; 40.18°N, −104.73°W, 1523 m asl), Fort

Collins-West (FCW; 40.89°N, -105.13°W, 1572 m asl), NREL-Golden (Golden; 39.74°N, -105.18°W, 1833 m asl), and CDPHE wind measurements at Weld County tower (WC-Tower; -104.73 W, 40.39 N, 1483 m asl). BAO, PAO, and WC-Tower are located north of Denver and close to the Wattenberg Gas Field in Weld County (Figure 1). Measurements of temperature, relative humidity, wind speed and direction at 10m, 100m, and 300m were recorded at BAO. Surface wind measurements from PAO and WC-Tower were used in this study. The planetary Boundary Layer (PBL) height was measured

and calculated at PAO, Fort Collins-West (FCW), and NREL-Golden using micro-pulse Lidar backscatter during the daytime (Compton et al., 2013).

### 2.2. WRF-Chem model

We used WRF-Chem 3.6.1 (Grell et al., 2005; Skamarock et al., 2008), a fully coupled online air quality and transport model,

to investigate the sensitivity of modeled PBL, winds, temperature, relative humidity, and ethane concentrations to different physical parametrizations and configurations. Figure 1 illustrates the location of the two nested domains and the underlying terrain map. We used one-way nesting (i.e., the outer domain ran independently of the inner domain). The outer domain has a 12 km × 12 km horizontal resolution, and the inner domain has a 4 km × 4km horizontal resolution. Both domains have 53 vertical levels with the domain top at 50 hPa (~11 layers below 1km). The outer domain is designed to capture the emission

from the Western US, and the inner domain includes Colorado and Utah. Sensitivity simulations start on 24 July 2014 and end on 18 Aug 2014. Tables and Figures

Table 1 shows a summary of the WRF-Chem configurations for this study, used in all sensitivity simulations. We selected the Regional Atmospheric Chemistry Mechanism chemistry using Earth System Research Laboratory (RACM-ESRL) (Stockwell et al., 1997) coupled to the Modal Aerosol Dynamics Model/Secondary Organic Aerosol Model (MADE/SORGAM).

RACM_ESRL (Kim et al., 2009) is an updated version of the RACM mechanism and includes 23 photolysis and 221 chemical reactions (Ahmadov et al., 2015). Chemical boundary conditions from Monitoring Atmospheric Composition and Climate reanalysis (MACC) (Inness et al., 2013) and model outputs from RAQMS (Natarajan et al., 2012; Pierce et al., 2007) were used as chemical boundary and initial conditions.



### 2.2.1.    WRF-Chem sensitivity tests

WRF-Chem provides users with a number of different dynamical, physical, and chemical schemes. Users can select schemes based on the physical properties of the domain of interest, goals of the study, and computational limitations. We evaluated the sensitivity of WRF-Chem to different physics options, such as the PBL parametrization, and configurations including daily re-initialization of meteorological fields, different meteorological initial and boundary conditions, and varying horizontal resolution. Table 2 shows details on the sensitivity runs and lists the meteorological and chemical boundary conditions used for each run. The naming system for the simulations is based on the different settings (e.g. simulation 4-MnERi represents the simulation number (4), PBL Scheme (MYNN), meteorological initial and boundary condition (ERA-interim), chemical initial and boundary condition (RAQMS), and daily re-initialization of meteorological fields (i)). Simulation ID in Table 2 has been used when discussing sensitivity tests in the paper.

An accurate simulation of air pollution is dependent on a precise description of transport processes, meteorological conditions, and the PBL height (PBLH) (Cuchiara et al., 2014; Hu et al., 2010). Transport of pollutants within the domain depends on turbulent motions and vertical mixing within the PBL. WRF-Chem (3.6.1) has eleven different PBL schemes to address the closure problem in the simulation of turbulent motions. In general, PBL schemes can be classified into two main groups; local and non-local. A local PBL scheme estimates the turbulent fluxes of heat, momentum, and moisture from local mean and gradient flux values. In a non-local PBL scheme, non-local fluxes can influence fluxes in each grid, hence they are expected to better capture large-size eddies in the simulation (Stull, 1988). We tested one non-local and two local PBL schemes to understand the sensitivity of the model to PBL parameterization in a domain with high elevation and complex terrain. We used Yonsei University (YSU) first order (Hong et al., 2006) as the non-local PBL scheme in the PBL1 (1-YFM) simulation. The local schemes used in PBL2 (2-MjFM) and PBL3 (3-MnFM) simulations were Mellor–Yamada–Janjic (MYJ) 1.5 order (2.5 level) (Janjic, 2001; Janjic et al., 2000) and Mellor-Yamada-Nakanishi-Niino (MYNN3) 3rd level (Nakanishi and Niino, 2009).

WRF-Chem is a mesoscale model and requires initial and boundary conditions from a larger-scale model. Usually, these initial and boundary conditions are taken from re-analysis products of larger-scale models optimized using assimilation techniques and observations. The choice of initial and boundary condition products can impact the model performance (Angevine et al., 2012; Saide et al., 2011). We tested two different meteorological initial and boundary conditions, European Reanalysis (ERA-interim) by European Center for Medium-Range Weather Forecasts (ECMWF) in Met5 (5-MnERi) simulation and NCEP's Global Forecast System (GFS) in Met6 (6-MnFRi) simulation. ERA-Interim reanalysis is produced with 80km by 80km horizontal and 6-hour temporal resolution (European Centre for Medium-Range Weather Forecasts (ECMWF), 2009), and NCEP FNL (final) operational global analysis is produced using GFS with 1-degree by 1-degree horizontal and 6-hour temporal resolution (National Centers for Environmental Prediction, National Weather Service, NOAA, 2000).



Simulations were performed for 24 days from 24 July 2014 to 18 August 2014. Initializing the meteorological fields in the simulation at the first time step with the larger-scale model values and running it for 24 days without any nudging will result in deviations from the larger scale re-analysis products. On the other hand, the lower resolution of the larger-scale models can lower the accuracy of WRF-Chem high-resolution simulations. To investigate this impact, we tested two different set-ups for

5 WRF-Chem. In Init4 (4-MnER) simulation, we initialized the meteorological fields at the first time step with larger-scale model values and ran the simulation freely for 24 days ("free run"). In Init5 (5-MnERi) simulation, the meteorological fields were re-initialized every day at 18 UTC (12pm local time) and run for the next 30 hours. The first 6 hours of the simulation (18 UTC to 00 UTC) were discarded to allow for the model to spin up. In this set-up, chemistry fields were recycled from previous cycles of simulations.

The sensitivity of the model to the horizontal resolution was examined by comparing the performance of the outer domain (12 km × 12 km) to the inner domain (4 km × 4 km) in Hor5 (5-MnERi) simulation. In one-way nesting, the outer domain runs independently of the inner domain; thus, comparing the performance of the outer and inner domains is valid.

### 2.3. Emission inventory

NEI-2011 version 2 is a bottom-up emission inventory of U.S. anthropogenic emissions. While we cannot expect the year 2011 inventory to fully represent the model year 2014, it was the only inventory available to the WRF-Chem user community at the time of this study. Emissions in this inventory are calculated based on fuel consumption, source activity, and emission factors reported by state, tribal, and local governing agencies (U.S. Environmental Protection Agency, 2015). WRF-Chem

provides a processed version of NEI-2011 to the users, which includes emission of 76 species (50 speciated VOC compounds, 19 PM2.5 aerosol species, and 7 primary species). NEI-2011 and emissions for only oil and NG sector in the NEI-2011 were provided to us by Dr. Stuart McKeen (NOAA Earth Systems Laboratory, Boulder, CO). The separate oil and NG emission information was used to conduct an additional sensitivity simulation with perturbed oil and NG emission, which we used to study the sensitivity of modeled ethane concentrations as well as concentrations of VOCs and CO to the oil and NG emission

sector. We used the Model of Emissions of Gases and Aerosols from Nature (MEGAN) for biogenic emission in all simulations (Guenther et al., 2012). Wildfire emissions were not included in the simulations, but this will have a negligible impact on the results as wildfires did not significantly influence the air quality in the NFR during the FRAPPÉ campaign (Valerino et al., 2017).

## 3. Results and Discussion

We start with an evaluation of the overall performance of all simulations and later provide a detailed discussion on the different sets of sensitivity simulations. To evaluate the sensitivity of WRF-Chem to different physical parametrizations, we compared the simulated meteorological variables, such as temperature, relative humidity, wind fields, and PBLH, with measurements.



27 July 2014 and 28 July 2014 were reported as Denver cyclone episodes (Dingle et al., 2016; Valerino et al., 2017; Vu et al., 2016), and neither simulation captured the cyclone pattern and enhancements accurately on these two days. Thus, we only included the period of 1 August 2014 to 15 August 2014 in our analysis to avoid skewing the results because of large model errors during the Denver cyclone episode. For quantitative comparison between the simulations, we used statistical measures including correlation coefficient (R), root mean square error (RMSE), mean average error (MAE), mean bias (MB), normalized mean bias (NMB), and index of agreement (IOA). Definitions of these metrics can be found in the supplement. We used NMB as a proxy for model sensitivity to quantify the impact of model configuration on different variables. Variability of NMB (calculated by subtracting minimum NMB from maximum NMB) in sensitivity tests can provide a range for uncertainties in the model cases independent of the model values. Table 3 includes the statistical measures for temperature and relative humidity in all the simulation tests at 100m altitude at BAO. Figure 2 compares the diurnal cycles of measured temperature, relative humidity, wind speed, and wind direction at 100m altitude at BAO with corresponding model values for all the simulation tests. Similarly, Table SM1 and SM2 includes statistical measures and Figure SM 1 shows diurnal cycles of temperature, relative humidity, wind speed, and wind direction at BAO 10m and 300m. All model simulations capture the overall daily cycle in temperature and relative humidity well (Figure 2 and Table 3). The variability across different sensitivity runs can be large, with modeled temperature varying by up to 6°C and the model-measurement NMB ranging from -3.9% to 11.1%. Relative humidity has larger variability among the simulations during nighttime compared to daytime. The NMB of relative humidity ranges from -29.7% to 52.6%.

Wind patterns vary significantly from daytime to nighttime. During the day, wind primarily blows from the east towards the Rocky Mountains with a slight southerly component. During the night, this pattern switches to predominantly westerly winds bringing cooler air to lower terrain. Wind measurements at the BAO at different altitudes (10m, 100m, 300m) can help us better understand the wind pattern at higher model levels. Table 4 includes mean and standard deviation of daytime and nighttime wind fields in the simulations and the observations at 100m. Results for the 10m and 300m level at BAO during 1 August 2014 to 15 August 2014 are included in table SM 3 and table SM 4, respectively. In addition to BAO, we investigated the wind sensitivity to physical parametrizations at two other sites that are close to oil and NG operations, WC tower and PAO (Figure SM 2). At BAO, higher wind speeds were measured at higher elevations which is captured by the model. Overall, all simulations show skill in capturing diurnal cycles of wind speed and direction with better agreement with observations for daytime (Table 4, Table SM 3, and Table SM 4). Complex local topography can cause localized transport patterns in the domain, which cannot be resolved at the model's 4 km x 4 km horizontal resolution. Pfister et al. (2017) discuss the impacts of the complicated wind patterns in the NFR and the limitations of WRF-Chem simulations in capturing the transport during FRAPPÉ campaign in details. To reduce the impact of localized influences on the sensitivity analysis we use airborne measurements which better represent the regional picture. Overall, the model runs show fairly good performance in capturing temperature, relative humidity, and wind fields, especially for daytime. A higher sensitivity to the physical parametrization was observed for nighttime.



Evaluation of modeled ethane concentrations with aircraft data provides information on the impact of different configurations on the transport of oil and NG emissions. Ethane is predominantly emitted from oil and NG production sites (Helmig et al., 2016; Xiao et al., 2008) and is a valuable chemical tracer to study the transport patterns of oil and NG emissions. Average ethane concentrations along the C130 morning and afternoon flights are shown in Figure 3. These data were acquired from the University of Colorado's CAMS instrument, details for which are discussed in Richter et al., 2015 . This plot limits the C130 observation to the NFR region (east of -105.2 longitude) to reduce transport errors, and it separates observations collected during 9:00 to noon (AM flights) and noon to 18:00 (PM flights) to account for the diurnal changes in PBLH. For this comparison, hourly model output has been interpolated to the time and location of each 1-minute average observation. Lower concentrations of ethane were measured during the PM flights compared to AM flights because of the higher PBLH and stronger vertical mixing in the afternoon. In all simulations, the ethane concentrations are under-predicted by up to 3.3 ppb (NMB ranges between -63% to -42%) for the C130 AM flights and up to 1.7 ppb (NMB ranges between -47.6% to -29.5%) for the C130 PM flights. Overall, measured ethane concentrations, absolute biases, and absolute NMBs are higher for C130 AM compared to C130 PM. However, the differences between variability in NMBs for C130 AM and C130 PM are small i.e., 21% and 18.1%.

Measurements from P3 spirals focus on smaller regions and can capture the impact of local emissions. Aerodyne Ethane-Mini spectrometer on P3 was used to measure ethane concentration (Yacovitch et al., 2014). Figure 4 compares the vertical distribution of measured ethane concentrations against the corresponding model values (interpolated to time and location of each 1-min average observation) for all the simulations at BAO and Platteville (PAO) spirals. Both sites are located close to oil and NG sources (Figure 1), however urban emissions from Denver region can reach BAO (Pfister et al., 2017a). Similar to C130 observations, we illustrate the morning and afternoon data separately. Fried, 2015 compared CAMS ethane measurements with the Aerodyne measurements during wing tip comparisons and the agreement was within 9%, corresponding to differences of less than 55 ppt. Mean concentrations of up to 18.6 ppb (SD 2.8 ppb) were measured by P3 aircraft, but these high values were not captured by the model and resulted in biases up to -14.9 ppb (NMB of -80.5%) at PAO spirals and biases up to -7.16 ppb (NMB of -57.8%) at BAO spirals. Similar to C130 flights, higher measured ethane concentrations, absolute biases, and NMBs are observed for P3 AM flights compared to PM flights. Variability in NMBs across simulations are greater in the PM spirals (42.8% at PAO and 57.3% at BAO) compared to AM spirals (36.5% at PAO and 31.3% at BAO).

While the model shows difficulty representing the absolute magnitude in the ethane concentrations, it represents the profile shape of ethane concentration well. The C130 flights covered a larger region with varying flight patterns across the NFR, thus less variability in the modeled ethane concentrations was observed compared to the P3, which flew a repetitive pattern and the repeated spirals over the key surface locations reflect a higher influence from localized emissions.



### 3.1.    Sensitivity to Planetary Boundary Layer Parametrization

We evaluated the sensitivity of WRF-Chem meteorological fields and ethane concentrations to a non-local (YSU) and two local (MYJ and MYNN) PBL schemes in PBL1, PBL2, and PBL3 simulation, respectively. Table 2 includes details on

simulation configurations. Temperature at BAO changes little between the different PBL schemes and the model agrees with observations (Figure 2). At all three altitudes, PBL1 has a small positive bias (errors less than 1ºC) while PBL2 and PBL3 have a small negative bias (errors less than 1ºC) (Table 3 and Table SM 1). Relative humidity differed slightly between local and non-local PBL parametrizations. PBL1 captured relative humidity well, especially at lower altitudes (mean bias of 0.38%, 1.47%, and 4.93% for 10m, 100m, and 300m respectively). PBL2 and PBL3 both over-predicted relative humidity at all

altitudes. The mean bias for PBL2 and PBL3 ranged between 11.12% to 14.78% and 6.61% to 9.55%, respectively.

At all altitudes of BAO, PBL1 predicted higher wind speeds than observed as well as PBL2 and PBL3. Figure 5 compares the 10m average wind speed (during 1-August to 11-August) in PBL1, PBL2, and PBL3 for daytime and nighttime. Higher daytime wind speed was predicted by PBL1 in the Colorado Eastern plains, especially north of Denver and close to oil and

NG operations (Figure 5). Wind direction does not vary significantly between PBL1, PBL2, and PBL3 at BAO tower and the model missed the southerly component of afternoon winds. Figure SM 2 shows the averaged diurnal cycle of wind speed and wind direction at WC Tower and PAO (sites close to oil and NG operation). At WC tower and PAO, PBL2 and PBL3 better captured the southerly component of afternoon winds compared to BAO.

Each PBL scheme in the WRF model uses different diagnostics to determine the PBLH. To have a consistent comparison of PBL height in the three simulations, we used the 1.5-theta-increase method to estimate PBL height. In this method, PBLH is the lowest altitude where the difference between minimum potential temperature and potential temperature is greater than 1.5 K (Hu et al., 2010; Nielsen-Gammon et al., 2008). Figure 6 shows the diurnal evolution of PBLH as calculated using the 1.5-theta-increase method in the simulations. Observed PBLH at the PAO, Fort Collins-West (FCW), and Golden-NREL sites

were retrieved from micro-pulse Lidar backscatter profiles (Compton et al., 2013), thus a quantitative comparison between model and measurement is not possible. However, a qualitative comparison of the PBLH growth in the model with measurements is valid. PBLH in the PBL1 simulation is greater than either PBL2, PBL3, or observations, and the bias is largest in the afternoon.

The high bias in temperature, wind speed, and PBLH in PBL1, non-local PBL scheme, suggests a strong vertical mixing that is more defined in the Colorado Eastern plains and close to the oil and NG activities. PBL2 and PBL3, local PBL schemes, predict cooler and moister climates and lower PBLH, which indicates less vertical mixing. This is consistent with previous works that compared local and non-local PBL schemes in the WRF model (Angevine et al., 2012; Hu et al., 2010). To evaluate the impact of vertical mixing intensity on the distribution of pollutants, we compared the vertical distribution of ethane



concentrations between the three simulations. Figure 7 shows the vertical cross section of ethane on 12-August-2014 at PAO for the three simulations. PBL1 distributed ethane higher into the atmosphere and, because of more dilution resulted in lower ethane concentration within the PBL. Similar impact on the vertical distribution of ethane was observed on other days and locations (not shown). Figure SM 3 shows, on average, up to 5 ppb higher surface ethane in simulations based on local PBL schemes (PBL2 and PBL3) compared to the simulation based on non-local PBL scheme (PBL1).

The comparison between C130 airborne measurements and modeled ethane across the NFR, as illustrated in Figure 3, shows biases between -2.5 ppb and -2.3 ppb for AM flights and between -1.7 ppb and -1 ppb for PM flights. Lower NMB variability (4%) was observed in the C130 AM with NMB ranging from -43.1% to -47.1% compared to C130 PM with NMB variability of 18% and NMB ranging from -29.5% to -47.6%. Similar to the C-130 comparison, the simulations did not capture the high ethane values measured during P3-BAO and P3-PAO spirals. The sensitivity of modeled ethane profiles to the PBL scheme is larger in P3 flights compared to C130 flights, with NMB variability of 14.1% ranging from -58% to -44% for PAO AM flights and NMB variability of 32.4% ranging from -37.3% to -69.7% for the PAO PM flight.

## 3.2. Sensitivity to re-initialization

We investigated the impact of daily initialization of meteorological fields on the model performance in capturing the transport of pollutants. For this, we conducted a sensitivity simulation (Init5) in which each daily cycle started at 18 UTC from ERA-interim meteorological fields and ran for 30 hours. In the comparison free-running simulation, Init4, we initialized the model at the first time step using the ERA-interim model and ran the simulation from 24 July 2014 to 18 August 2014 freely. Figure 2 shows an up to 3°C bias in nighttime temperature in Init5, but good agreement with the measured temperature during the day. Init4 showed better skill in capturing nighttime temperature compared to Init5, but predicted the lowest daytime temperature among all the simulations with a bias up to -3°C. On average, the NMB of the temperature at BAO100m is between 8.6% in Init5 compared to -6.0% in Init4 (Table 3), which is the largest variability in NMB temperature across the simulations. Similar to the temperature, relative humidity showed a strong sensitivity to re-initialization. Init4 predicted the highest relative humidity, with NMB of 39.2% and Init5 predicted the lowest relative humidity, with a NMB of -26.5% among the simulations at BAO 100m (Table 3). Nighttime wind direction at BAO (Figure 2), PAO, and WC tower (Figure SM 2) had a strong southerly component in Init4 compared to Init5 and observations. In addition, Init4 predicted higher wind speeds compared to Init5.

When compared to C130-AM ethane concentrations (Figure 3), Init4 predicted the lowest ethane concentration (a bias of -3.3 ppb and NMB of -63%) among all the simulations. The ethane bias is ~-2.5 ppb and NMB is -47.9% in Init5 during C130-AM. Concentrations during the C130-PM flights showed a weak sensitivity to re-initialization with NMB ranging from -37.8% (Init4) to -40.1% (Init5). For the P3-BAO and P3-PAO spirals in both AM and PM flights, Init4 had the lowest ethane values compared to all the other simulations and compared to observations (Figure 4). This resulted in the largest NMB variability



across the simulations. During PAO AM, NMB ranges between -80.5% for Init4 and -53.2% for Init5 (NMB variability of 27.3%) and during PAO PM, NMB ranges between -72.9% for Init4 and -30.0% for Init5 (NMB variability of 43.9%).

### 3.3. Sensitivity to meteorological initial and boundary condition

5  We tested the performance of changing the meteorological initial and boundary conditions by comparing simulations using ERA-Interim (Met5) with simulations using NCEP-FNL (Met6). As was done for Met5, we initialized meteorological fields with the re-analysis fields every day allowing for a 6-hour spin-up. Figure 2 and Figure SM1 indicate that the performance of the two simulations is comparable in capturing temperature and relative humidity. Met5 had slightly higher temperature during the night and lower relative humidity compared to Met6 and compared to measurements.

Comparison of ethane measurements by the C-130 and P3 aircraft with Met5 and Met6, shown in Figure 3 and Figure 4, respectively, also reflects an overall low sensitivity of the model performance to meteorological initial and boundary condition for both AM and PM flights. High sensitivity was observed during P3-PAO PM flight with ethane NMB variability of 23.9% where Met5 had a bias of -2.6 ppb (NMB of -30%) and Met6 had -4.7 ppb (NMB of -53.9%). The highest differences between

Met5 and Met6 prediction of ethane occurred on 27 July 2014 and 28 July 2014. These two days were reported as Denver cyclone episodes. A Denver cyclone is a mesoscale vortex circulation pattern driven by the terrain is common during the summer which can affect the regional air quality in the NFR (Dingle et al., 2016; Valerino et al., 2017; Vu et al., 2016). Small misplacement in the location the Cyclone can lead to large differences in the modeled concentrations. During this event, both simulations missed the high enhancement of ethane, however ERA-interim simulation predicted higher values of ethane during

this event. Figure SM 4 compares the averaged surface ethane in Met5 and Met6 during Denver cyclone episodes and a regular day. During the Denver cyclone episode (Figure SM 4-a), Met5 predicted higher concentration of surface ethane compared to Met6, but the deviation between Met5 and Met6 is much smaller in a regular day (Figure SM 4-b).

### 3.4. Sensitivity to Horizontal resolution

The two nested domains in simulation Hor5 had a horizontal resolution of 12 km × 12 km (coarse) and 4 km × 4 km (fine). The one-way nesting method was used to prevent any feedback from the higher resolution inner domain on the outer domain. Thus, these two domains run independently. To compare the impact of horizontal resolution, we compared the performance of the coarse domain with the fine domain in the same simulation (5Hor). Temperature and relative humidity did not show significant sensitivity to the horizontal resolution at BAO, as neither did surface winds at BAO (Figure 2), PAO, and WC

tower (Figure SM 2). At 100m and 300m altitudes at BAO, the coarse domain predicted higher wind speed compared to the fine domain.



Averaged ethane concentrations along the C130 flights (Figure 3) do not vary significantly with horizontal resolution. However, higher differences are observed for the P3 spirals. This might be due to the C130 flights covering a larger area and, in parts, averaging out the impact of horizontal resolution, whereas the P3 spirals capture small-scale transport patterns in the domain more effectively. For the P3 spirals (Figure 4), the ethane NMB during BAO PM is +11.6% for the fine domain and -

21.6% for the coarse domain. These values are -30% and -55.1% during PAO PM flights, respectively.

### 3.5.    Oil and NG emission in the NFR

We assessed the performance of the model in capturing oil and NG emissions by focusing on ethane, which is mostly emitted from oil and NG emission sources, and on species with multiple emission sources such as CO and other VOCs. To investigate

the contribution of oil and NG emissions to NFR air quality, we ran two additional simulations: in the one, the emissions are based on the NEI-2011 as provided (base simulation or Em7), in the other we doubled the oil and NG emissions (perturbed simulation or Em8).

Figure 8 shows the C130 PM measurements limited to altitudes below 1500m and Figure 9 displays scatterplots of measured

to modeled species concentrations for the same data. High values of ethane were measured in the vicinity of oil and NG facilities which were not captured by the model (as shown in Figure 8a) resulting in low biases (Figure SM 5a). As can be expected, the simulated ethane concentrations show a high sensitivity to changes in the oil and NG emissions (Figure SM 6). The highest sensitivity was observed for measurements taken over regions close to oil and NG sources, such as the P3-PAO spirals. Ethane biases between Em7 and Em8 varied from -9.4 ppb to -1 ppb (NMB from -50.8% to -5.5%) during P3-PAO

AM, and from -2.7 ppb to +2.8 ppb (-31.2% to +31.8%) during P3-PAO PM. Doubling oil and NG emissions lowered the absolute bias during the AM flights (NMB from -50.8% to -5.5%) , but resulted in an overestimation of ethane during the PM flights (NMB from -31.2% to +31.8%). One possible reason for the difference between AM and PM biases might be an incorrect representation of the diurnal variation of ethane emission rates in the NEI-2011. An inverse modeling technique, as will be subject of further studies, can be used to calculate optimal scaling factors for hourly ethane emissions with the goal to

minimize the discrepancies between model and measurement.

CO is mostly emitted from combustion processes and is released from many different source sectors. The measured CO enhancements over Denver and oil and NG facilities, shown in Figure 8b, do not vary greatly between Em7 and Em8. Biases along the C130 flight tracks (Figure SM 5b) show an over-prediction of CO over Denver and west of Denver and an under-

prediction over the oil and NG facilities. The scatterplot in Figure 9b reflects an overall low bias in modeled CO. Doubling oil and NG emissions in Em8 only marginally increased the slope of the regression line indicating a low sensitivity of CO in the NFR to oil and NG emissions. This suggests that the source of the low bias in CO likely is related to other source categories and/or the model boundary conditions.




In the RACM chemical mechanism, alkanes such as propane, n-butane, isobutane, and acetylene (ethyne), and alcohols such as methanol and ethanol are lumped under the "HC3" group (Stockwell et al., 1990). We compared the simulated HC3 concentrations with the sum of measured chemicals in the HC3 group during C130 flights. Similar to ethane, the highest values of HC3 were measured over oil and NG facilities (Figure 8c). These enhancements were not captured in the model and resulted

in low model biases (Figure SM 5c). Comparison of measured HC3 with modeled values from Em7 and Em8, Figure 9c, confirms the low bias of HC3 and shows some increase in the slope of the regression line in Em8 albeit less pronounced compared to ethane.

Toluene and benzene are lumped together in the RACM chemistry under "TOL" (Stockwell et al., 1990). We compared

simulated TOL with the sum of toluene and benzene concentrations observed during the C130 flights. The transport sector is a strong source of toluene and benzene in the NFR as well as oil and NG activities. TOL enhancements were observed over oil and NG facilities and Denver with higher values associated with oil and NG emissions (Figure 8d). The model did not capture the enhancements in regions influenced by oil and NG emissions, but well captured TOL values over Denver (Figure SM 5d). TOL showed very low sensitivity to perturbed oil and NG emissions as shown in Figure 9d. TOL emissions from oil

and NG sector in the emission inventory used in this study (NEI-2011) were very low thus doubling oil and NG emissions did not increase TOL in the Em8. Similar to toluene and benzene, xylene enhancements were measured over oil and NG facilities and Denver. Model underestimated xylene enhancements over oil and NG activities and overestimated these enhancements over Denver. Em8 with doubled oil and NG emissions showed very similar performance to Em7 which indicates low emission rates of xylene from oil and NG sector in the NEI-2011 (not shown).

Figure 10 illustrates the HC3 to TOL ratio measured along the C130 PM and the corresponding model values. Figure 10a shows oil and NG influenced points with enhanced measured ethane (concentrations greater than 2 ppb). HC3 to TOL ratios in oil and NG influenced locations show inconsistency between measured and modeled ratios which was improved in the Em8. However, doubling oil and NG emission still resulted in underestimations of HC3, TOL, and their ratios in this region. Figure

10b shows urban influenced points with low measured ethane (concentrations less than 2 ppb). Modeled HC3 to TOL ratios in the urban influenced locations agreed well with the measurements but showed an underestimation in the HC3. One reason for this offset can be leakage from the NG distribution system which was not captured in the model. As expected, HC3 to TOL ratio in this region did not show large sensitivity to oil and NG emissions.

The results suggest that HC3, toluene, benzene, and xylene from oil and NG sector are significantly underestimated in the NEI-2011. The low model bias for these species is more pronounced compared to the low model bias in ethane. The inconsistency between these biases implies that the NEI-2011 emission ratios might need to be changed and HC3, toluene, benzene, and xylene oil and NG emissions would need to be increased by a larger factor than ethane.



## 4.    Conclusion

We used WRF-Chem to understand the sensitivity of pollutant transport at a high horizontal resolution to different model configurations with the focus on oil and NG emissions. By conducting a range of different sensitivity simulations, we assessed the variability of meteorological variables such as temperature, relative humidity, and wind fields as well as of ethane concentrations (used as a tracer for the oil and NG sector) to different model configurations and parameterizations. The overall daily cycle of temperature and relative humidity was captured well in the simulations with NMB ranging from -3.9% to 11.1% in temperature and from 29.7% to 52.6% in relative humidity. All simulations showed good skill in capturing daytime wind fields but showed higher biases for nighttime wind speeds.

Table 5 summarizes the mean and NMB for ethane concentrations from C130 and P3 airborne measurements below 2000m agl and the corresponding model values for all sensitivity tests. Significant underestimation of ethane in all simulations– especially in regions close to oil and NG activities – with biases up to -14.9 ppb (NMB up to -80.5%) suggest that the emission inventory used (NEI-2011) under-predicts oil and NG emissions. NMB variability was used as a proxy for variability in the model performance caused by model configurations. NMB of the near-surface ethane concentration for aircraft flight patterns across sensitivity simulations varied by up to 57.3% for P3-BAO, by up to 42.8% for P3-PAO and by up to 21.1% for C130 flights. The lower NMB variability during C130 flight can be due to the larger area coverage by this aircraft during the FRAPPÉ campaign and the irregular flight patterns. P3 spirals, covering smaller regions within the domain during repetitive flight patterns, focused more on the local emissions and smaller scale transport patterns and captured a larger ethane sensitivity to model configurations. The largest sensitivity occurred in the initialization test (comparing daily re-initialization with free-run simulation) with ethane NMB variability up to 57.3%, followed by the horizontal resolution test (comparing horizontal resolution of 12 km × 12 km with 4 km × 4 km) and the PBL parametrization test (comparing local with non-local PBL schemes) with ethane NMB variability up to 33.3% and 32.4%, respectively.

We investigated the performance of the model in capturing oil and NG emission in the NFR by comparing measured ethane, CO, lumped alkanes (HC3), lumped toluene and benzene (TOL), and xylene to corresponding modeled values and assessed the changes in the model performance when doubling oil and NG emissions. The model showed under-prediction of ethane with the original inventory and a strong sensitivity of ethane concentrations to oil and NG emissions. Doubling oil and NG emissions resulted in an improvement during AM flights and an overestimation of ethane during the PM flights which suggests possible incorrect representation of the diurnal variation of ethane emission rates in the NEI-2011. The model tends to overestimate CO over the Denver region and underestimates CO over the oil and NG region. Low sensitivity of CO to oil and NG emissions indicates that CO in the region is predominantly emitted from sources other than oil and NG. Enhancements of HC3, TOL, xylene over oil and NG facilities were not fully captured in the model and resulted in low biases. Doubling emissions from oil and NG emissions improved the model performance in capturing HC3, but still resulted in a low model bias. Although high values of TOL and xylene were measured over oil and NG facilities, the model did not capture these



enhancements in either the simulations with base NEI-2011 emissions or doubled oil and NG emissions. The inconsistency between the sensitivity of ethane, HC3, benzene, toluene, and xylene to the increase in oil and NG emissions and mismatch between VOC ratios in the model and measurement suggest that oil and NG emission rates in the NEI-2011 need to be scaled differently for these species. VOC ratios in the measurements can be used to update these ratios in the emissions inventory.

The presented results reflect the challenges that one is faced with when attempting to improve emission inventories by contrasting measured with modeled concentrations, either through simple direct comparisons or more advanced methods, such as inverse modeling. Any uncertainties that arise from the model configuration will translate into the derived emission constraints, and it is important to be aware of the uncertainties resulting from different model setups. The WRF-Chem

simulations and knowledge gained from this study will be used to support inverse modeling studies aimed to improve estimates of emission from oil and NG sector in the NFR.

**Acknowledgements**

The authors would like to acknowledge the FRAPPÉ and DISCOVER-AQ science team. The authors acknowledge the use of WRF-Chem version 3.6.1 and the NCAR Command Language (UCAR/NCAR/CISL/TDD, 2017). We acknowledge Stuart

McKeen (NOAA) for providing NEI-2011 emission inventory, Bradley Pierce (NOAA) for providing RAQMS model outputs for chemical initial and boundary conditions, and Ravan Ahmadov (NOAA) for help with conducting WRF-Chem simulations. We thank Gordon Pierce (CDPHE/APCD) and Erick Mattson (CDPHE/APCD) for providing wind data at CDPHE site, Daniel Wolfe (NOAA) for providing meteorological data at BAO tower, William Brune (Pennsylvania State University) for providing meteorological data at PAO site. We further acknowledge Teresa Campos (NCAR/ACOM) for C130 CO measurements, Lisa

Kaser (NCAR) for C130 PTR-MS VOC measurements, Eric Apel (NCAR/ACOM) for C130 TOGA VOC measurements, Don Blake (UC Irvine) for aircraft VOC WAS measurements. The University of Iowa group activities are partly funded by the Regional Scale Modeling in Support of KORUS-AQ: Improving Predictions of Dynamic Air Quality using Aircraft, Ground Networks, and Satellite Data (Award No: NNX15AU17G) and Regional-Scale Analysis of Gas and Aerosol Distributions and the Development of Emissions in Support of the NASA SEAC4RS Mission (Award No: NNX12AB78G).

This work was supported by computational resources provided by the University of Iowa.



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



**Tables and Figures**

**Table 1. Summary of basic WRF-Chem configuration**

| Category | Selected option |
|---|---|
| **Horizontal resolution** | 12km and 4km |
| **Vertical resolution** | 53 layers (11 within the lowest 1km) |
| **Microphysics** | Morrison double-moment scheme |
| **Land Surface** | 5-layer thermal diffusion |
| **Shortwave radiation** | Goddard shortwave |
| **longwave radiation** | RRTMG scheme |
| **Gas-phase chemistry** | RACM-ESRL |
| **Biogenic emission** | MEGAN |



**Table 2. Summary of WRF-Chem configurations for sensitivity tests designed for this study**

| Test | Sim. ID | Sim. Name | PBL Scheme | Met IC & BC | Chem IC & BC | Init. | Emiss. |
|---|---|---|---|---|---|---|---|
| PBL | PBL1 | 1-YFM | YSU (Y) | NCEP-FNL (F) | MACC (M) | Free run | NEI2011 |
| | PBL2 | 2-MjFM | MYJ (Mj) | NCEP-FNL (F) | MACC (M) | Free run | NEI2011 |
| | PBL3 | 3-MnFM | MYNN3 (Mn) | NCEP-FNL (F) | MACC (M) | Free run | NEI2011 |
| Initialization | Init4 | 4-MnER | MYNN3 (Mn) | ERA-interim (E) | RAQMS (R) | Free run | NEI2011 |
| | Init5 | 5-MnERi | MYNN3 (Mn) | ERA-interim (E) | RAQMS (R) | re-init (i) | NEI2011 |
| Met IC & BC | Met5 | 5-MnERi | MYNN3 (Mn) | ERA-interim (E) | RAQMS (R) | re-init (i) | NEI2011 |
| | Met6 | 6-MnFRi | MYNN3 (Mn) | NCEP-FNL (F) | RAQMS (R) | re-init (i) | NEI2011 |
| Horizontal resolution | Hor5 | 5-MnERi | MYNN3 (Mn) | ERA-interim (E) | RAQMS (R) | re-init (i) | NEI2011 |
| | Hor5-12km | 5-MnERi-12km | MYNN3 (Mn) | ERA-interim (E) | RAQMS (R) | re-init (i) | NEI2011 |
| Emission Inventory | Em7 | 5-MnERiMeg | MYNN3 (Mn) | ERA-interim (E) | RAQMS (R) | re-init (i) | NEI2011 + Megan |
| | Em8 | 7-MnERiMeg-2OnG | MYNN3 (Mn) | ERA-interim (E) | RAQMS (R) | re-init (i) | NEI2011 (doubled oil & NG) + Megan |





**Table 3. Summary of model performance in capturing temperature (T) and Relative Humidity (RH) at BAO 100m during 1 to 15 August 2014.**

| T(C)-100m | PBL | | | Met IC and BC | | Initialization | | Horizontal resolution | |
|---|---|---|---|---|---|---|---|---|---|
| | PBL1 | PBL2 | PBL3 | Met5 | Met6 | Init4 | Init5 | Hor5 | Hor5-12km |
| Mean Model | 22.18 | 21.15 | 21.52 | 23.92 | 23.20 | 20.70 | 23.92 | 23.92 | 23.90 |
| Mean Obs | 22.01 | 22.014 | 22.01 | 22.01 | 22.01 | 22.01 | 22.01 | 22.01 | 22.01 |
| R | 0.85 | 0.83 | 0.81 | 0.81 | 0.84 | 0.63 | 0.81 | 0.81 | 0.82 |
| RMSE | 1.86 | 2.07 | 2.01 | 2.74 | 2.17 | 3.07 | 2.74 | 2.74 | 2.72 |
| MAE | 1.40 | 1.72 | 1.65 | 2.18 | 1.64 | 2.46 | 2.18 | 2.18 | 2.10 |
| MB | 0.17 | -0.86 | -0.5 | 1.90 | 1.19 | -0.31 | 1.90 | 1.90 | 1.89 |
| NMB | 0.8% | -3.9% | -2.3% | 8.6% | 5.4% | -6.0% | 8.6% | 8.6% | 8.6% |
| IAO | 0.92 | 0.89 | 0.89 | 0.82 | 0.89 | 0.76 | 0.82 | 0.82 | 0.83 |
| **RH (%)-100m** | **PBL1** | **PBL2** | **PBL3** | **Met5** | **Met6** | **Init4** | **Init5** | **Hor5** | **Hor5-12km** |
| Mean Model | 43.74 | 51.79 | 48.88 | 31.06 | 38.51 | 58.90 | 31.06 | 31.06 | 31.52 |
| Mean Obs | 42.27 | 42.27 | 42.27 | 42.27 | 42.27 | 42.30 | 42.27 | 42.27 | 42.27 |
| R | 0.69 | 0.59 | 0.53 | 0.52 | 0.52 | 0.44 | 0.52 | 0.52 | 0.58 |
| RMSE | 11.90 | 16.33 | 14.67 | 16.69 | 13.63 | 25.90 | 16.69 | 16.69 | 16.00 |
| MAE | 9.21 | 13.47 | 12.31 | 12.79 | 10.28 | 21.17 | 12.79 | 12.79 | 11.99 |
| MB | 1.47 | 9.52 | 6.61 | -11.21 | -3.76 | 16.63 | -11.21 | -11.21 | -10.75 |
| NMB | 3.5% | 22.5% | 15.6% | -26.5% | -8.9% | 39.2% | -26.5% | -26.5% | -25.4% |
| IAO | 0.82 | 0.70 | 0.68 | 0.63 | 0.70 | 0.52 | 0.63 | 0.63 | 0.66 |




**Table 4. Summary of model performance in capturing wind speed and direction at BAO 100m during Aug 1-15, 2014**

| | | PBL | | | Met IC & BC | | Initialization | | Horizontal Res. | |
|---|---|---|---|---|---|---|---|---|---|---|
| **Day - 100 m** | | **PBL1** | **PBL2** | **PBL3** | **Met5** | **Met6** | **Init4** | **Init5** | **Hor5** | **Hor5-12km** |
| **Wind Speed** | Mean Model | 3.84 | 3.40 | 2.70 | 2.87 | 3.19 | 3.77 | 2.87 | 2.87 | 2.76 |
| | STD Model | 2.14 | 2.26 | 1.57 | 1.57 | 1.80 | 2.86 | 1.57 | 1.57 | 1.45 |
| | Mean Obs | 3.22 | 3.22 | 3.22 | 3.22 | 3.22 | 3.22 | 3.22 | 3.22 | 3.22 |
| | STD Obs | 2.02 | 2.02 | 2.02 | 2.02 | 2.02 | 2.02 | 2.02 | 2.02 | 2.02 |
| **Wind Direction** | Mean Model | 62.90 | 64.05 | 66.76 | 33.86 | 59.61 | 55.92 | 33.86 | 33.86 | 41.11 |
| | STD Model | 48.79 | 63.44 | 56.30 | 73.10 | 75.90 | 74.77 | 73.10 | 73.10 | 67.74 |
| | Mean Obs | 117.84 | 117.84 | 117.84 | 117.84 | 117.84 | 117.84 | 117.84 | 117.84 | 117.84 |
| | STD Obs | 71.06 | 71.06 | 71.06 | 71.06 | 71.06 | 71.06 | 71.06 | 71.06 | 71.06 |
| **Night - 100 m** | | **PBL1** | **PBL2** | **PBL3** | **Met5** | **Met6** | **Init4** | **Init5** | **Hor5** | **Hor5-12km** |
| **Wind Speed** | Mean Model | 4.69 | 4.06 | 3.57 | 4.02 | 4.41 | 4.87 | 4.02 | 4.02 | 4.73 |
| | STD Model | 2.34 | 2.78 | 2.47 | 2.45 | 2.32 | 2.88 | 2.45 | 2.45 | 3.15 |
| | Mean Obs | 3.42 | 3.42 | 3.42 | 3.42 | 3.42 | 3.42 | 3.42 | 3.42 | 3.42 |
| | STD Obs | 1.81 | 1.81 | 1.81 | 1.81 | 1.81 | 1.81 | 1.81 | 1.81 | 1.81 |
| **Wind Direction** | Mean Model | 114.12 | 268.45 | 349.75 | 331.38 | 292.24 | 155.59 | 331.38 | 331.38 | 303.89 |
| | STD Model | 97.13 | 89.35 | 86.75 | 87.28 | 77.12 | 85.20 | 87.28 | 87.28 | 85.11 |
| | Mean Obs | 233.09 | 233.09 | 233.09 | 233.09 | 233.09 | 233.09 | 233.09 | 233.09 | 233.09 |
| | STD Obs | 70.62 | 70.62 | 70.62 | 70.62 | 70.62 | 70.62 | 70.62 | 70.62 | 70.62 |





**Table 5. Ethane mean and standard deviation (SD) from C130 and P3 BAO and PAO airborne measurements below 1500m agl and the corresponding model values**

|  |  |  | C130 - NFR | | P3 - BAO | | P3- PAO | |
|---|---|---|---|---|---|---|---|---|
|  |  |  | AM | PM | AM | PM | AM | PM |
|  | **OBS** | **Mean (ppb)** | 5.22 | 3.49 | 12.39 | 4.90 | 18.56 | 8.66 |
| **PBL** | **PBL1** | **Mean (ppb)** | 2.97 | 1.83 | 8.51 | 3.85 | 7.79 | 2.62 |
|  |  | **NMB (%)** | -43.1 | -47.6 | -31.3 | -21.4 | -58.0 | -69.7 |
|  | **PBL2** | **Mean (ppb)** | 2.97 | 2.36 | 9.11 | 4.53 | 7.93 | 5.43 |
|  |  | **NMB (%)** | -43.1 | -32.4 | -26.5 | -7.6 | -57.3 | -37.3 |
|  | **PBL3** | **Mean (ppb)** | 2.76 | 2.46 | 8.67 | 4.90 | 10.40 | 4.20 |
|  |  | **NMB (%)** | -47.1 | -29.5 | -30 | 0 | -44.0 | -51.5 |
| **Init.** | **Init4** | **Mean (ppb)** | 1.93 | 2.17 | 5.23 | 2.66 | 3.62 | 2.35 |
|  |  | **NMB (%)** | -63.0 | -37.8 | -57.8 | -45.7 | -80.5 | -72.9 |
|  | **Init5** | **Mean (ppb)** | 2.72 | 2.09 | 7.46 | 5.47 | 8.68 | 6.06 |
|  |  | **NMB (%)** | -47.9 | -40.1 | -39.7 | 11.6 | -53.2 | -30.0 |
| **Met IC & BC** | **Met5** | **Mean (ppb)** | 2.72 | 2.09 | 7.46 | 5.47 | 8.68 | 6.06 |
|  |  | **NMB (%)** | -47.9 | -40.1 | -39.7 | 11.6 | -53.2 | -30.0 |
|  | **Met6** | **Mean (ppb)** | 3.03 | 1.92 | 7.00 | 4.46 | 7.90 | 3.99 |
|  |  | **NMB (%)** | -42.0 | -45.0 | -43.5 | -9.0 | -57.3 | -53.9 |
| **Hor. Res** | **Hor5** | **Mean (ppb)** | 2.72 | 2.09 | 7.46 | 5.47 | 8.68 | 6.06 |
|  |  | **NMB (%)** | -47.9 | -40.1 | -39.7 | 11.6 | -53.2 | -30.0 |
|  | **Hor5-12km** | **Mean (ppb)** | 2.60 | 1.98 | 5.67 | 3.84 | 5.68 | 3.89 |
|  |  | **NMB (%)** | -50.2 | -43.3 | -54.2 | -21.6 | -69.4 | -55.1 |
| **Emiss** | **Em7** | **Mean (ppb)** | 2.76 | 2.16 | 7.59 | 5.26 | 9.13 | 5.96 |
|  |  | **NMB (%)** | -47.1 | -38.1 | -38.6 | 7.3 | -50.8 | -31.2 |
|  | **Em8** | **Mean (ppb)** | 5.07 | 3.90 | 14.57 | 10.1 | 17.54 | 11.41 |
|  |  | **NMB (%)** | -2.9 | 11.7 | 17.6 | 106.1 | -5.5 | 31.8 |





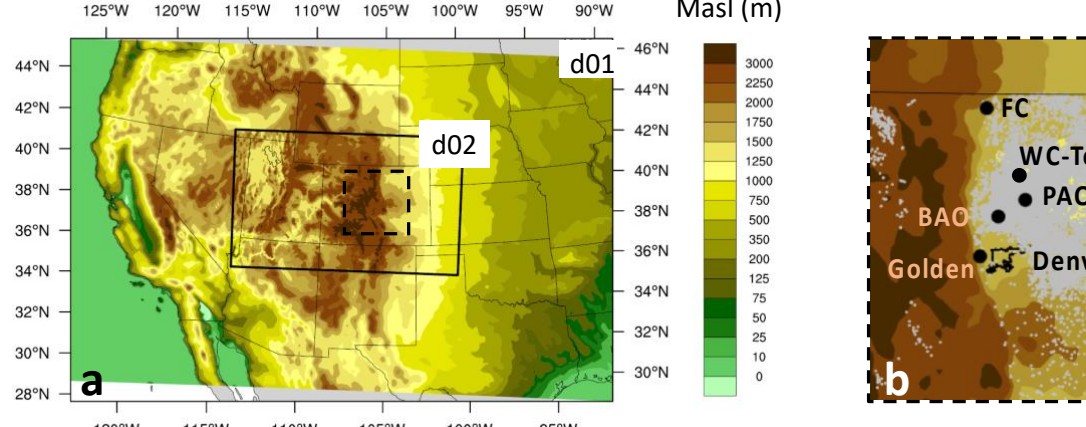

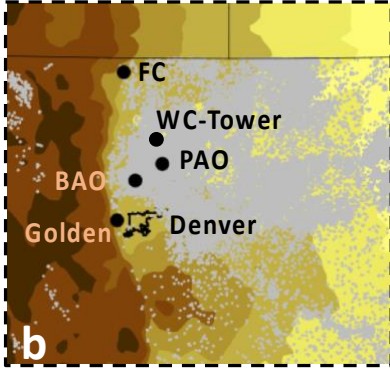

**Figure 1. Terrain map of the WRF-Chem domains and location of observation sites. Grey dots show the location of permitted wells (http://cogcc.state.co.us/).**





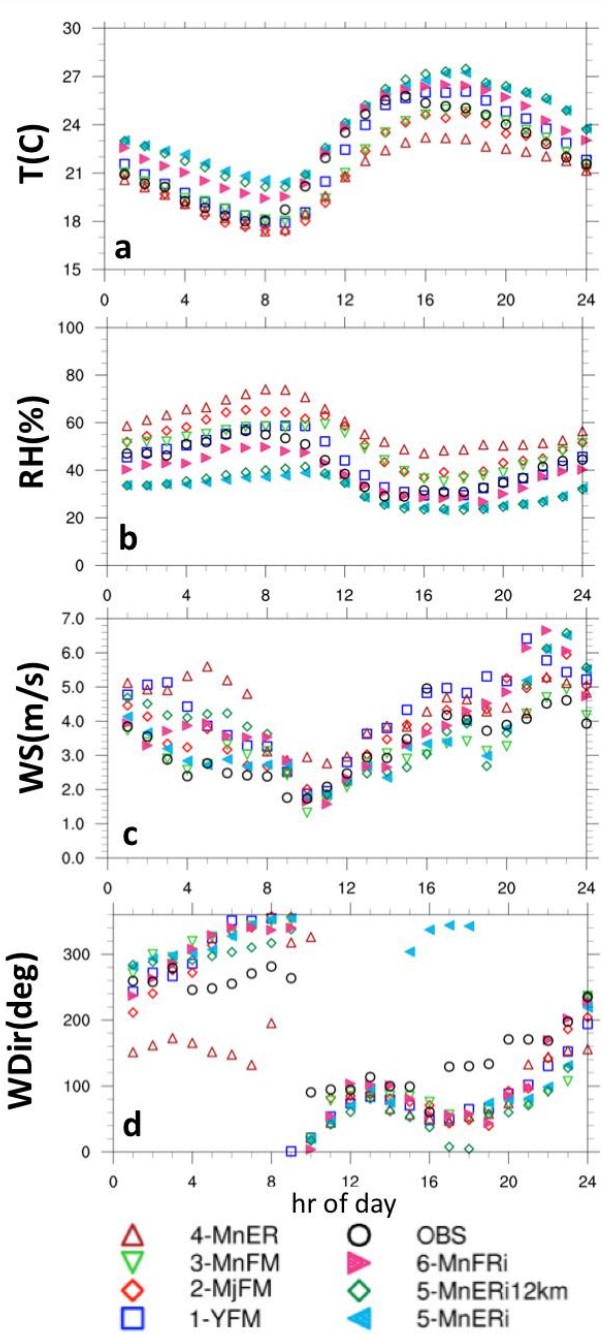

**Figure 2. Average diurnal cycle of temperature (a), relative humidity (b), wind speed (c) and wind direction (d) for all tests and observation at BAO 100m. Averages are calculated for Aug 1 to 15, 2014.**



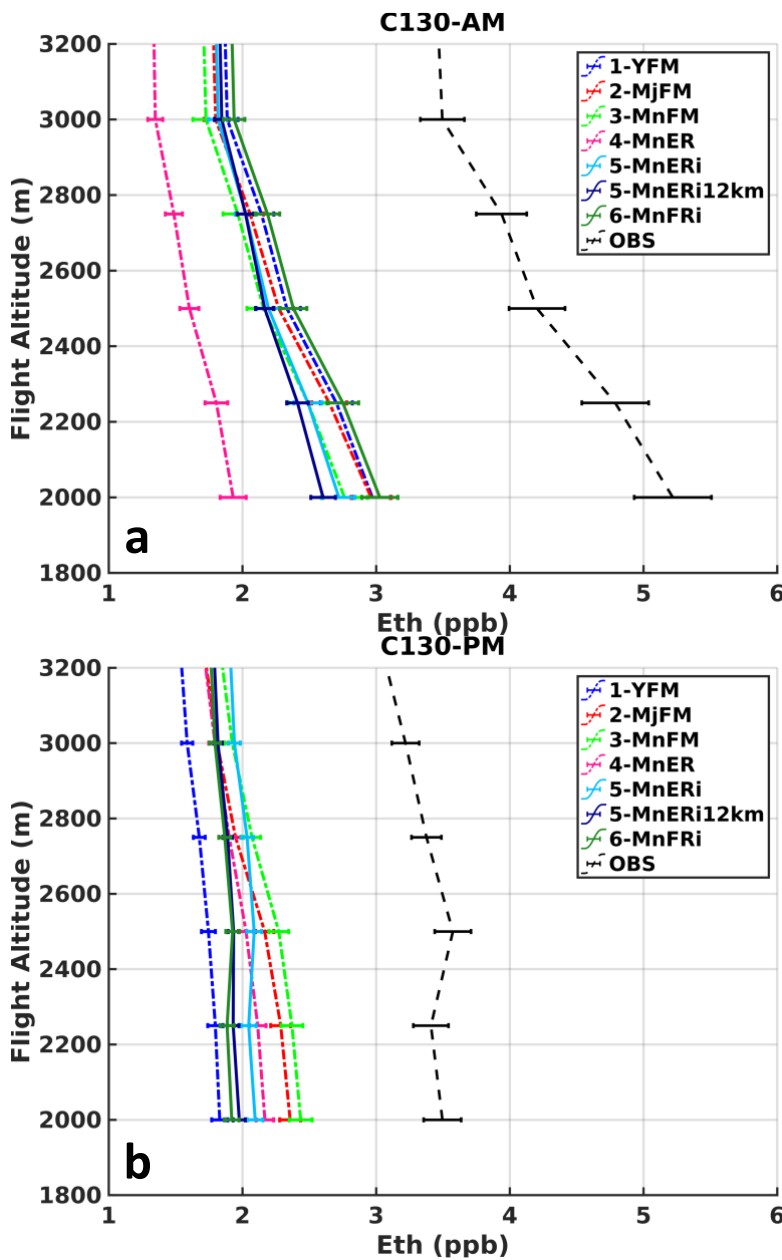

**Figure 3. Vertical distribution of simulated and measured ethane in the NFR area separated by the flight time. (a) C130-AM 9am to noon observation and the corresponding model values. (b) C130-PM noon to 6pm observation and the corresponding model values. Error bars represent the standard error.**



**Figure 4. Vertical distribution of ethane at PAO (a and b) and BAO (c and d) site measured during P3 spiral flights and the corresponding model values. Flights are separated by the flight time. a and c show P3-AM that include 9am to noon observation and the corresponding model values. b and d show P3-PM that include noon to 6pm observation and the corresponding model values. Error bars represent the standard error.**



**Figure 5. Wind speed at 10m captured by different PBL schemes averaged from 1-August to 11-August 2014.**





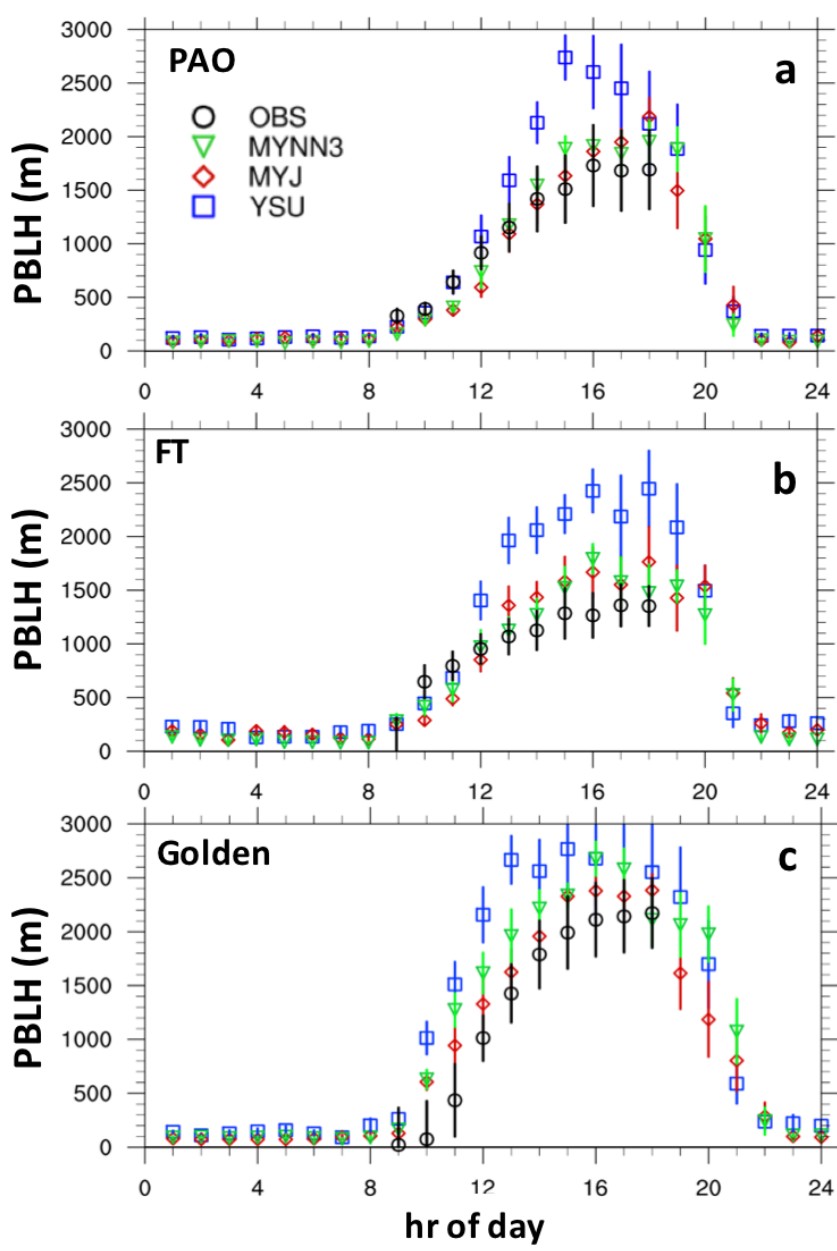

**Figure 6. Diurnal evolution of PBL in MYJ, MYNN3, and YSU schemes at PAO (a), Fort Collins (b), and Golden-NREL (c) sites. PBLH was measured using micro-pulse Lidar backscatter profiles during the daytime. Error bars represent the standard error.**





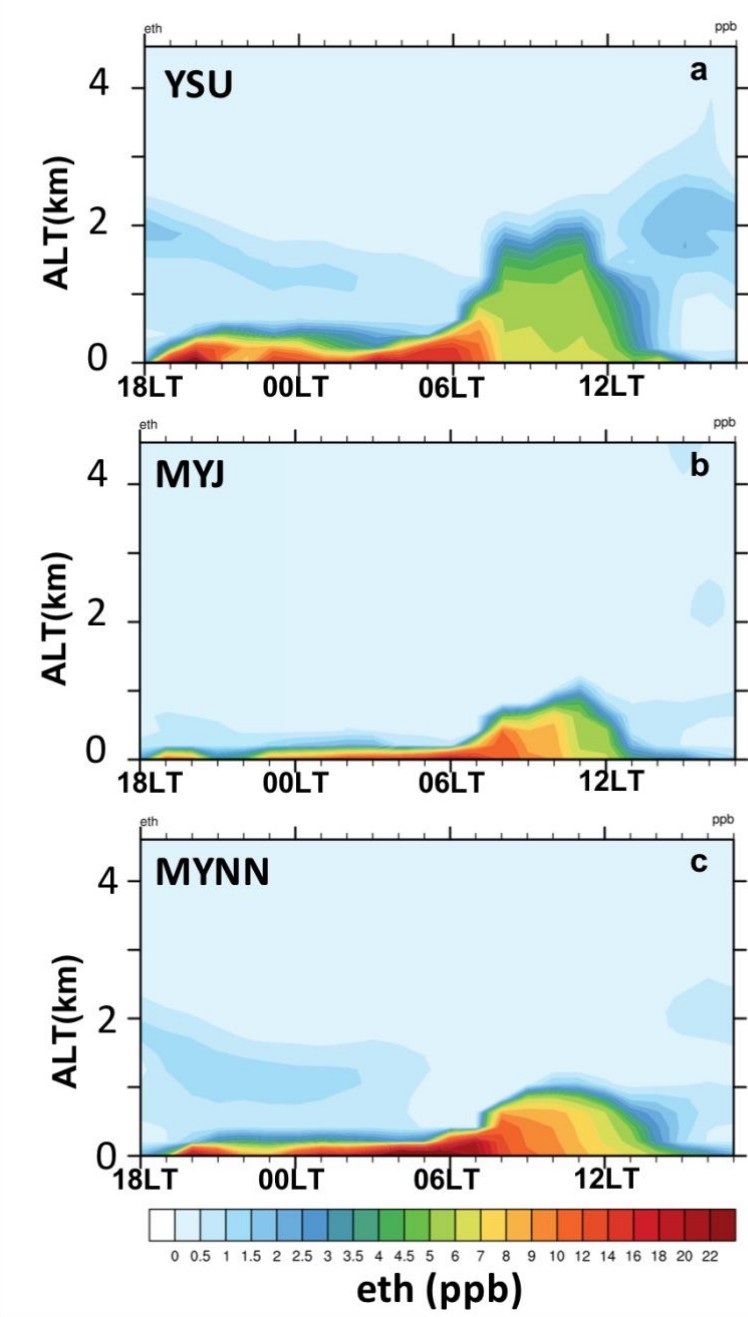

**Figure 7. Impact of PBL parametrization on vertical distribution of ethane. Curtain plots show ethane in YSU (a), MYJ (b), and MYNN (c) simulation at PAO on August-12-2014.**





**Figure 8. Mean ethane (a), CO (b), HC3 (c), and TOL (d) measurements along the C130 PM flights limited to measurements below 2000m agl and grids with more than 4 measurement points. Outline of the Denver county and location of BAO and PAO sites are marked on the underlying terrain map.**





**Figure 9. scatter plot of measured vs. corresponding model values of ethane (a), CO (b), HC3 (c), and TOL (d) along the C130 PM flights limited to measurements in the NFR and below 2000m. Red diamonds represent the Em7 (base emissions) and blue circles represent Em8 (perturbed emissions). Red and Blue lines show the best fit using least square linear regression method for Em7 and Em8, respectively.**





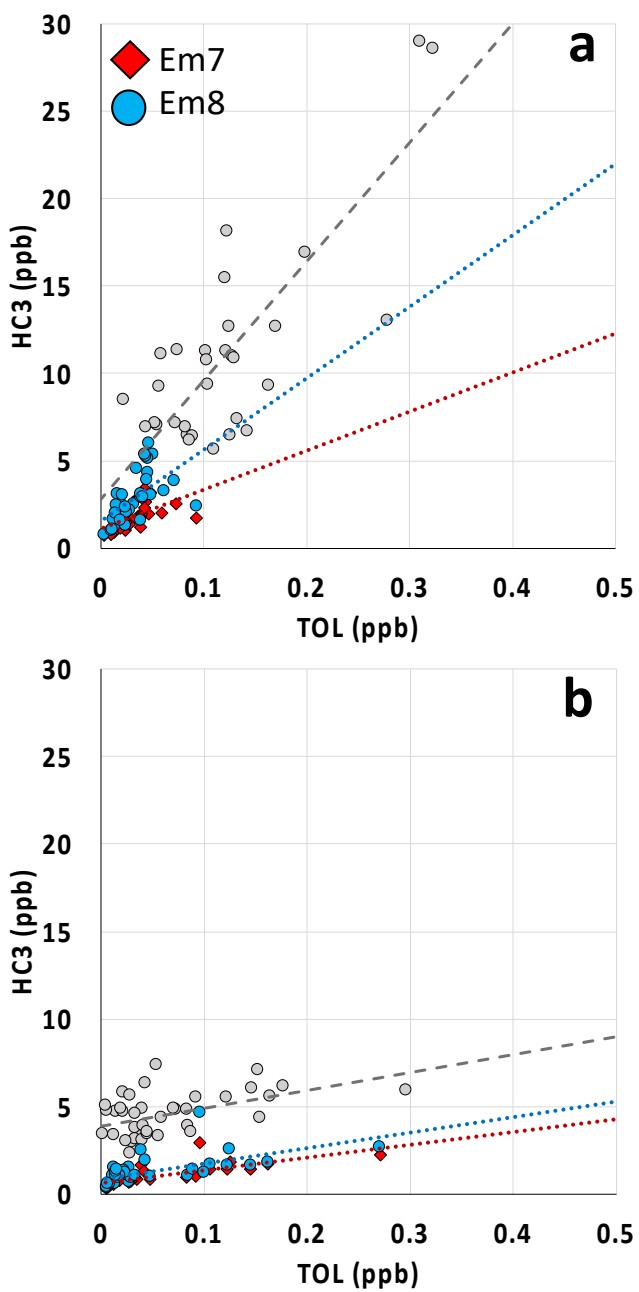

**Figure 10. scatter plot of HC3 vs. TOL along the C130 PM flights limited to measurements in the NFR and below 2000m altitude. Plot (a) shows measurements with ethane greater than 2ppb and the corresponding model values. Plot (b) shows measurements with ethane less than 2ppb and the corresponding model values. Grey circles represent measurements, red diamonds represent the Em7 (base emissions), and blue circles represent Em8 (perturbed emissions). Grey, Red and Blue lines show the best fit using least square linear regression method for observations, Em7 and Em8, respectively.**