# Peer review of "Impacts of physical parametrization on prediction of ethane concentrations for oil and gas emissions in WRF-Chem"

_Atmospheric Chemistry and Physics, 2018_

## Referee Comment (RC1) · Anonymous Referee #1 · 25 May 2018

**Review on: "Impacts of physical parametrizations on prediction of ethane concentrations for oil and gas emissions" by Maryam Abdi-Oskouei et al.**

In this article the authors test the influence of different configuration of the WRF-Chem model on the modeled ethane concentration. More specifically, they test three boundary layer schemes, two different dynamical boundary conditions, a scaling of the emission inventory, the effect of a free running simulation versus a re-initialised one and the impact of the horizontal resolution (12 km vs. 4 km). Investigation of the influence of different model configurations on the simulation results is an important issue, specificially for chemistry simulations where ensemble simulations can not be performed due to limitations in computing power and storage space. Unfortunately, the article stays at the surface of such investigations, solely comparing the differences of the simulations. Neither the authors go into the detail where these differences come from, nor do they draw any conclusions on the configuration, which should be chosen in future simulations.

This article requires at least major revisions. From my point of view a re-submission after a thorough revision would be the better way, as the article needs to provide much more substantial information to the reader.

The following points are ordered by appearance in the article, not by importance:

Major points:

Sect. 2.2:

- With the horizontal resolution of 4 km x 4 km the smaller domain is well within the gap between the horizontal resolution where convection parametrisations are fully applicable and the convection resolving scale. The information, which convection parametrisations are still active in the 4 km domain (all or only deep convection) needs to be added to the model description. Note, that the influence of the choice of the convection parametrisation on the simulation result needs to be discussed (at least in the section where the influence of the horizontal resolution is investigated).

- Many informations on the chemical initial and boundary conditions (IC/BC) is missing: In which time interval are new BC data applied? Are the IC/BC data provided for this specific time interval/date? If yes, how do they fit to the two different dynamical boundary conditions used?

- How many species are included in the used chemical mechanism?
Sect. 3.:

- The title of the section is wrong. So far it provides only a listing of differences between the sensitivity simulations (results), but almost always neither an explanation for the differences is provided, nor is discussed, what the consequences for future simulations are (discussion).

- Use of the supplement for figures and tables:
  All in all I very much appreciate to provide additional information in the supplement. Anyhow, in your case I got the feeling that the article should be shortend by putting tables important for the discussion into the supplement, while desirable extra information which would be well suited for the supplement is missing. More precise:

    – Table SM4 (maybe SM3) should become part of the main paper.

    – Figure 2 might provide a nice overview over all simulations, but it does not provide enough information for the comparison of the respective sensitivity simlations. Simply because of the number of lines (different symbols) not all simulations are identifyable. Here the supplement could be used to provide a figure 2 like figure individually for each of the tested parametrisations (one for the PBL schemes, one for (re-)init, one for IC/BC, one for horizontal resolutions and on for the different emission scenarios).

    – Figures SM1 / SM2: some as for Fig. 2 add individual panels for those comparisons referring to SM2 (PBL and re-init). Maybe add reference to SM 2 also for the other tested parametrisations?

    – Figure 3 and Figure 4: Same as for Fig. 2. Use the supplement to provide the plots individually for each sensitivity study, where the lines are hardly distinguishable otherwise.

    – Figure 3/4: dashed and solid lines are not distinguishable in the legend.
- – Why are the tables for the model performance only provided for BAO station? It would be nice to be able to compare the so far provided statistics to the other stations listed in the introduction.
- – Tab. (4 / SM3 / SM4 ) ( 3 / SM2) Why do you show observations in a row? In this way you have to repeat the value for the observations for each simulation. Just show obs in an additional column! In this way you could easily combine Tables 4, SM 3 and SM 4 and Tables 3 and SM 2 in one landscape table, respectively.

- p. 7, 26./27.: Here I disagree, not "all" simulations show better skill at daytime. Better change "all simulations" to "most simulations".

- p.8, l. 15 / l. 29: I am missing any information why the AM and PM flights are different and any suggestions, why the modeled concentrations are off by such a hugh amount. If these points would be sufficiently discussed in the following subsections, it would be ok to refer to the following analysis, but so far, they are not discussed in the further analysis either.

- Sect. 3.1:
  - – as already mentioned,the difference between the PBL simulations hardly visible in Fig. 2. It would be good to have a figure showing only the three simulations and the observations in the supplement.
  - – Why do you use different chemical initial and boundary conditions for the PBL sensitivity studies (MACC) in contrast to all other sensitivity studies (Table 2)? This courses additional differences you never discuss.
  - – p.9, l.16: Cite Figure SM1.
  - – Fig. 6: Article Text states "Fort-Collins-West (FCW)" , figure shows "FT".
  - – p.9, l.26: For a non-specialist for boundary height measurements: Why is a quantitative comparison not possible?

- – end of section: What are the consequences? Does the choice of PBL scheme matter or not? Which PBL scheme will you pick in future?

- Sect. 3.2:

  - – You change here from simulations driven by NCEP and MACC data, to simulations driven by ERAinterm and RAQMS data. Some information, why not the same driving data is used is missing here.
  - – p.10, l.28.: "in addition, Init 4 predicted higher wind speeds compared to Init5." And compared to Observations?
  - – p.10, l.28.: Any suggestion, which simulation is more realistic?
  - – p.10, l. 30/31 and 34 : Why does Init4 show the by far lowest ethane concentrations for C130-AM and all P3 flights? I thought the chemistry is not re-initialised in the same way as the meteorology?

- Sect. 3.3:

  - – The evaluation how much the simulation changes with respect to the meteorological initial and boundary data is meaningless without a comparison of the differences in the driving data themselves. How close are the reanalyses of ERA-Interim and NCEP-NFL to the observations in the area of interest?
  - – p. 11, l. 7-9: as far as I can deduce the correct lines from Fig. 2 and SM 1, both simulations overestimate temperature and underestimate moisture, where the ERA-Interim simulations is more off than the NCEP simulation.
  - – p.11, l. 14/15: I do not understand, I thought the Denver cyclone was not part of the analysis (top page 7). If this is just another "feature" and not the explanation of the huge differences in Met6 simulation for P3-PAO add this explanation and indicate by a line break, that you are talking about something new.

– p. 11, l.19/20: Why do the simulations miss the event?

– p. 11, l.22 / SM4: Please show the absolute values (at least of one of the simulations) as well, otherwise absolute differences do not provide the full picture. This Figure should be part of the main paper. If you want to discuss the Denver cyclone, figure SM4 should be part of the main paper. But in this case, the analyses should go beyond the mere stating of the differences.

Sect. 3.4:

– p. 11, l. 27: No, the domains are not run "independently" as the 4x4 $km^2$ domain depends on the 12x12 $km^2$ domain.

– p.11, l. 28./29.: What about temperature and relative humidity at PAO and WC tower?

– p. 11, l. 28 /31: The different labels are hard to detect in the Figure 2 / SM2.

– p. 11, l. 31. This is only true at nighttime (Tab. 4 / SM 4). Cite the tables in the text.

Sect. 3.5:

– p. 12, l. 16: According to caption, Fig. 8 shows measurements. How can it be seen, that the model does not match the measurements? This is seen from Fig. SM 5, therefore this figure belongs in the article, not in the supplement. The phrase "as shown in Fig. 8a" is very irritating. It belongs in front of the "which", i.e., after the statement about high values of ethane.

– as the comparison of the two emission scenarios is the topic of this subsection, SM 6 also belongs into the main paper. However, Fig. 9 could be part of the Supplement.

– p.13, l.18/19: I do not think that you need an indicator for low emission rates. These are input by the emission inventory, thus this is aprior knowledge.

– Discussion of Fig. 10: Say more about the best fit lines (meaning of differences in intercepts and slope).

– p. 13, l. 31/32: Where do I see these low biases (cite respective table / figure)

• What about cross evaluation? What changes with different chemical IC/BC? What is the influence of the PBL scheme for the higher emission szenario?

Conclusions:

• p. 14, l.10-22: In this section "NMB" and "NMB variability" is falsely used synonymously. It might be true that the inter-simulation variability of NMB might be usable "as a proxy for variability in the model performance caused by model configuration". But the percentages listed below are only the NMBs and not the inter-simulation variability of the NMB.
However, this paragraph again only lists (new) results (not even a discussion). Therefore it does not belong in the conclusion section.

• p. 14, l. 24 - p.15, l. 4: This section stays very in-concrete. All the species discussed here have a medium range lifetime. Nevertheless, the aspect of the influence of chemical initial and boundary conditions is completely left out.

• For me the final conclusion is missing: Based on this study, what has to be taken into account by setting up a WRF-Chem simulation? What would be the best WRF-Chem setup?

Minor points:

• From my point of view WRF-Chem is not a Chemical Transport Model (CTM). The term "transport" includes the fact, that the chemical substances are transported,

but they do not feed back to the dynamics. WRF-Chem is a fully featured regional atmospheric chemistry model (RACM) which very well includes e.g., radiation and cloud feedback. Therefore I recommend to omit the term CTM with respect to WRF-Chem. I know that whole communities use this term wrongly, but this did not change the misconception of it.

- As this article is very WRF-Chem specific, it would be apropriate to name the model in the title of the article, e.g. "Impacts of physical parametrizations on prediction of ethane concentrations for oil and gas emissions a case study with WRF-Chem v3.6.1" or 'Impacts of physical parametrizations of WRF-Chem (v3.6.1) on prediction of ethane concentrations for oil and gas emissions"

- Sect. 2.3: Add more information on how the emissions of 76 species are mapped to the species of the chemical mechanism used.

- p.7, ll. 5-17: add the information, that these results will be discussed in more detail in the following subsections.

- p.7, ll. 19-34: add the information, that these results will be discussed in more detail in the following subsections.

- p. 10, l. 10: cite Table 5.

- p. 10, l. 11: cite Fig. 4.

- S1: Sums over i without any "i"s in the equations? What is "n"? IOA is introduced (also in the article) but never referenced.

Typos & Co:

- p.4, l. 26: remove "Tables and Figures"

- p.5, l. 8: Did you consciously choose an abbreviation (4-MnERi) which is not included in Table 2 ?

- p.5, l. 9: The abbreviation of the PBL scheme is (MYNN3)

- Through out the article (and in the figures) inconsistent abbreviations for Fort-Collings West are used (FC, FCW ...). This should be uniform.

- Fig. 5: What is the meaning of the line at 41 N ?

- p. 9. l. 11: Cite SM 3/4 at end of sentence.

- p. 9. l. 11: Fig. 5 does not "compare" anything, it "visualises", "shows", "depicts" ...

- p. 9, l. 30 / 31: use full sentences: "PBL1, non-local PBL scheme" → " PBL1 using a non-local PBL scheme".

- p. 11. l. 20: A figure does not "compare" anything, it "visualises", "shows", "depicts" ...

- p. 12, l. 14, Fig. 8: Limited to 200agl or 1500 m ?

- Fig. 8.: If these are measurements, what does "grids with more than 4 measurement points" mean?

- Fig. 10: Improve caption. The second sentence sound as if model vs. measurements are plotted.

- Fig. 10 / 9: give parameters of best fit line.

- Tab. 5: Is the NMB for PBL3 P3-BAO really 0 ?

- SM2: caption: state that "surface winds" are shown.

- Table SM4: omit line breaks for init5

- Fig. SM3: What is surface ethane: ethane in the lowest model layer?
* * *

---

## Referee Comment (RC2) · Anonymous Referee #2 · 16 Jul 2018

The manuscript 'Impacts of physical parametrisation on prediction of ethane concentrations for oil and gas emissions' by Maryam Abdi-Oskouei and co-authors describes a model sensitivity study applying WRF-CHEM to air pollutant simulations in the Denver, Colorado, area. The study focuses on the effect on tracer transport related to oil and gas extraction. The manuscript is well written and overall well structured. For the most part, the applied methods are appropriate and constructive. However, some additional clarifications concerning the aim of study and the drawn conclusions are required before the manuscript can be published.

**Major comments**

Aim, approach and conclusions: The aims of the presented study are somewhat twofold. 1) evaluate the chemistry-transport model and 2) draw some preliminary conclusions on the emissions from oil and gas extraction in the Northern Front Range Metropolitan area (NFRMA). In general, I agree with the authors that a transport model needs to be validated before it can be used for inverse modelling purposes (here emission estimation). However, these two subjects are somewhat mixed together in the study, since one main evaluation parameter of the model system is the ethane concentration, which strongly depends on the emissions, which one finally would like to determine. The manuscript would gain in significance if these two subjects (transport model performance and emissions from oil and gas) could be separated more more clearly in the presentation of the results. Instead of evaluating model performance for ethane another model tracer (e.g. carbon monoxide) should be given more attention during the evaluation part of the manuscript. Since carbon monoxide is largely unrelated to the oil and gas emissions and its emissions are otherwise relatively well know, one should be able to distinguish, which model configuration is best suited for tracer transport simulations. Only afterwards, the less well known ethane emissions should be discussed.

In this context another question arises concerning the application of the full chemistry version of WRF-CHEM in the sensitivity runs. A passive tracer setup (as available in WRF-CHEM) would have been sufficient to carry out the sensitivity runs as well. Only the discussion including shorter lived VOCs really grants running the complete model. This is more a comment for future sensitivity simulations than for the current manuscript. However, since the complete chemistry model was run, it would also be interesting to have a look at a secondary pollutant (e.g., ozone) and how it reacts to the emission variations in the final sensitivity experiments. Most likely ozone observations were available from all flight campaigns and additionally surface sites. The model-

observation mismatch for ozone might deliver another hint towards the suspected underestimation of emissions from gas and oil extraction.

Finally and coming back to the beginning, the conclusions of the paper should give more emphasise to the finally selected model configuration. Which is the best configuration, which parameters were most important to change, and why does the change make sense? Without such a recommendation (although not necessarily valid for other study areas or periods) the whole presentation of sensitivity results remains of no avail for the reader who wants to find concrete hints for setting up his/her own simulations. Right now the conclusion only summarises all sensitivity simulations together, although there were clear performance difference for some of the runs (PBL1 vs PBL3, free-run vs re-ini, etc). These are important details of the model setup that should be more easily accessible when browsing the paper.

**Minor comments**

Page 2, Line 9: From the context and my own knowledge of emissions from the oil and gas sector it remains unclear why NOx emissions should play a major role in the extraction process. Is it due to fossil-fuel operated machinery (compressors, pumps, etc) or due to flaring? In the discussion of the model results there is no more mentioning of NOx either. Was it not observed on-board the aircraft?

P3, first paragraph: Others have used WRF-CHEM already for 'inverse modelling' of emissions from oil and gas extraction. The study by Barkley et al. (2017), which used WRF-CHEM for CH4 emission estimation (although not with full chemistry), should be mentioned as well. Either when mass balance approaches to estimate emissions are discussed or more general in the discussion of the results.

Subsection 2.1: The presentation of the observations is lacking any reference to the applied measurement techniques, data quality, etc. Some of this information is given

later when the results are presented, but it should actually be presented in this subsection. For the more involved measurements (VOC concentrations) it would also be nice to learn a bit about observation uncertainties before comparing with model results.

P4,L30: The kind of VOC lumping used in RACM should be mentioned. Is ethane treated explicitly? How many chemical species are present?

P5,L21 and elsewhere: Since it was mentioned above (P5,L10) that the simulation ID according to table 2 would be used throughout the text to identify the sensitivity simulations, I see no need to give the simulation name here as well. All these abbreviations will only confuse. All figures also contain a mix of names and identifiers that are not according to the ID. This should be unified, so that the reader only has to follow one set of abbreviations.

P6, 1st paragraph: What is the rational for this re-initialisation? The model domain is relatively small. Shouldn't the BCs dominate anyway? In this context it should also be mentioned why no meteorological data assimilation was run for WRF, which probably would have reduced the spread in the sensitivity tests significantly.

P7, L5: 'average error'. Should this be absolute error, which would actually fit the definition in the supplement.

P7, L6: Although the supplement provides the definition of the comparison parameter 'index of agreement', it would be nice to give a short interpretation of its benefits and expected value range for a well performing simulation. Furthermore, values of IOA are presented in a table, but never discussed in the text. Something that is true for other comparison parameters as well. If they are shown in the table a short description and interpretation in the text should also be given.

P7, L9: I feel like one could start a new subsection here (after 'model values') that deals with the general results. Everything before is an introduction to the results, now the first real results are shown.

[Figure]

P7, L25: Maybe I missed this somewhere else, but at which altitude were the measurements at WC tower and PAO taken?

P8, L12 and elsewhere in manuscript: Values of NMB are discussed. But are they shown anywhere in the tables or figures? If yes a cross-reference to the figure would be useful, if not this should also be mentioned (e.g. adding 'not shown'). Otherwise the reader will try to confirm this information in one of the figures.

P8, L31ff: It is argued that the model 'represents the profile shape [...] well.' I don't agree with this completely and for all cases. For the afternoon observations, where C130 and BAO observations actually show relatively low values at the bottom of the profile, most model runs still show a smoothly increasing concentration towards the surface. What is the potential reason for the observed profile shapes and what does the model miss here?

P9, L26: It is argued that a 'quantitative comparison between model and measurement [of PBLH] is not possible'. I don't agree. It depends on the situation, but during the day the agreement between a lidar-derived PBLH and one derived from temperature profiles should actually be pretty good as long as you have an aerosol-laden PBL, which is most likely the case in the American South-West. See also Collaud Coen et al. (2014).

Figure6: I suggest to extend this figure for all other sensitivity simulations as well. Next to wind speed, PBLH is the most important dispersion parameter on local to regional scale. The use of the same turbulence scheme in all other sensitivity runs will not necessarily lead to the same PBL heights. For example the two different BC runs are also likely to result in different PBL structures.

P10,L7ff: This paragraph should include a short description of the changes in profile shape for the different PBL runs. The interesting part is that with PBL2 and PBL3 and although these runs are supposed to produce more realistic PBL heights, the increase in PBL concentrations compared to PBL1 is opposite to what the observations suggest

(Fig 3 and 4). That is: afternoon concentrations increased from PBL1 to PBL2 and PBLE3 below a certain altitude, but the resulting profile shape is then even less in line with the observations.

P11,L14ff: In the beginning of the results section it is mentioned that the Denver cyclone period was removed from the analysis. Why is it presented here again? Either this part should be moved to the beginning of the results section when justifying why the phase was excluded from the analysis or it should be mentioned here again which period is really contained in the figures.

Section 3.5: Megan biogenic emissions are mentioned in the Table but not in the text. What is their significance for this sensitivity run and overall for this study? Did Megan predict significant emissions of ethane or HC3? Probably not!

P12,L27f: Not surprisingly, the measurements did not differ for the two different simulations! Should it be 'simulated CO'?

P13,L21ff: In the discussion of Figure 10 it should be mentioned that for the longer-lived HC3 one should/could have removed a background concentration. The presentation is still valid if only the slopes/ratios are discussed and background did not change over the domain of interest. The latter should be commented on.

P13,L26f: Following up on the last comment, it most likely is not recently emitted HC3 that leads to elevated levels but background levels. The alkanes contributing to HC3 are relatively long-lived and the HC3 concentration at the domain boundaries is most likely a few ppb. This could be easily checked in the model fields. In contrast TOL is relatively short-lived, so BC concentration are probably close to zero.

**Technical comments**

P2,L25: 'Thousand Cubic Feet' probably supposed to be 'Million Cubic Feet'. Also, I

am not sure if ACP allows US customary units.

P9,L22 and elsewhere: 'Fried, 2000' is not the correct formatting. Please refer to the ACP guidelines.

**References**

Barkley, Z. R., Lauvaux, T., Davis, K. J., Deng, A., Miles, N. L., Richardson, S. J., Cao, Y., Sweeney, C., et al.: Quantifying methane emissions from natural gas production in north-eastern Pennsylvania, Atmos. Chem. Phys., 17, 13941-13966, doi: 10.5194/acp-17-13941-2017, 2017.

Collaud Coen, M., Praz, C., Haefele, A., Ruffieux, D., Kaufmann, P., and Calpini, B.: Determination and climatology of the planetary boundary layer height above the Swiss plateau by in situ and remote sensing measurements as well as by the COSMO-2 model, Atmos. Chem. Phys., 14, 13205-13221, doi: 10.5194/acp-14-13205-2014, 2014.

---

## Author Comment (AC1) · 24 Sep 2018

*Response to Reviewers:*

**Impacts of physical parametrization on prediction of ethane concentrations for oil and gas emissions in WRF-Chem**

Maryam Abdi-Oskouei[1], Gabriele Pfister[2], Frank Flocke[2], Negin Sobhani[2], Pablo Saide[3], Alan Fried[4], Dirk Richter[4], Petter Weibring[4], and James Walega[4], Gregory Carmichael[1]

[1] Center for Global and Regional Environmental Research (CGRER), University of Iowa, Iowa City, Iowa, USA
[2] National Center for Atmospheric Research (NCAR), Boulder, Colorado, USA
[3] Department of Atmospheric and Oceanic Sciences, University of California Los Angeles (UCLA), Los Angeles, California, USA
[4] Institute of Arctic and Alpine Research, University of Colorado, Boulder, CO, USA

*Correspondence to*: Maryam Abdi-Oskouei (maryam-abdioskouei@uiowa.edu)

Dear Dr. Heini Wenli,

We thank you and the reviewers for the insightful comments, which have substantially improved the manuscript. We have revised the manuscript and addressed all comments raised by the reviewers. A version of the manuscript with highlighted modifications has been uploaded. Below you find our responses to the reviewer's comments. The reviewer's comments are shown in blue *italic*, our responses are shown in plain text, and the modified section of the manuscript is shown in **boldface**.

We do respect the comments raised by reviewers that there is a lack of conclusion on the best WRF-Chem configurations for this region. We highlighted the configurations we used for further emission sensitivity analysis in the conclusion section. However, we wanted to point out that the main goal of this paper is not to find an optimal WRF-Chem configuration but to quantify the sensitivity of modeled fields to different model configurations with significant implications for e.g. inverse modeling studies. We clarified this in the revised manuscript. Overall, changing one WRF-Chem model configuration may improve the performance of the model in some aspect of atmospheric chemistry and transport and may adversely impact another aspect. Thus, the "best configuration" can depend on the goal of the study. We have updated some of the figures and tables following the reviewers' comments to reduce clutter and convey information more effectively.

We appreciate your time and comments and look forward to your decision.

Best Regards,
Maryam Abdi-Oskouei, on behalf of all co-authors

Reviewer 1

General Comments

*R1:*
*In this article the authors test the influence of different configuration of the WRF-Chem model on the modeled ethane concentration. More specifically, they test three boundary layer schemes, two different dynamical boundary conditions, a scaling of the emission inventory, the effect of a free running simulation versus a re-initialised one and the impact of the horizontal resolution (12 km vs. 4 km). Investigation of the influence of different model configurations on the simulation results is an important issue, specifically for chemistry simulations where ensemble simulations can not be performed due to limitations in computing power and storage space. Unfortunately, the article stays at the surface of such investigations, solely comparing the differences of the simulations. Neither the authors go into the detail where these differences come from, nor do they draw any conclusions on the configuration, which should be chosen in future simulations.*
*This article requires at least major revisions. From my point of view a re-submission after a thorough revision would be the better way, as the article needs to provide much more substantial information to the reader.*

Response:
We highly appreciate the thorough review and helpful comments, which helped improve the quality of our manuscript. We revised the paper based on suggestions and comments. We have added more analysis and more detailed discussions throughout the text. A detailed reply to individual comments can be found below.

Major Points:
Sect. 2.2:
*R1:*
*With the horizontal resolution of 4 km x 4 km the smaller domain is well within the gap between the horizontal resolution where convection parametrisations are fully applicable and the convection resolving scale. The information, which convection parametrisations are still active in the 4 km domain (all or only deep convection) needs to be added to the model description. Note, that the influence of the choice of the convection parametrisation on the simulation result needs to be discussed (at least in the section where the influence of the horizontal resolution is investigated).*

Response:

We appreciate the point. We used the Grell-Freitas convective parametrization in all the simulations discussed in the paper. We have added information in text and in Table 1 regarding this scale-aware convective parametrization.

We have compared using Grell-Freitas convective parametrization with resolved convection in the inner (4 km x 4 km) domain but did not see significant impact on meteorology or ethane transport when we resolved convection in the inner domain compared to using a parameterized scheme. We did not include this analysis in the paper which focuses on physical parametrizations and set-ups with larger impacts on meteorological variables and ethane fields for in this study. Figure 1 shows measured and simulated diurnal cycles of meteorological variables at BAO averaged from August 1 to 15, 2014. The performance of the simulation with the Grell-Freitas convective parametrization is very similar to the simulation with resolved convection especially for temperature and RH (Similar results for 10m and 300m). Modeled ethane values along the flight paths are also very close for both simulations (Figure 2)

Text:

**The Morrison double-moment scheme was selected as the microphysics option and Goddard shortwave (Chou and Suarez, 1999) and RRTMG longwave radiation schemes (Iacono et al., 2008) were used as shortwave and longwave radiation parametrizations, respectively. The Grell-Freitas convection scheme (Grell and Freitas, 2014) was used as convective parametrization for both outer and inner domain. The inner domain falls into the "grey-scale" which means many of the assumptions used in convective parametrization will no longer be valid at this resolution. Grell-Freitas convection scheme is a stochastic scale dependent convective parametrization based on the method proposed by Arakawa et al. (2011) and is designed for domains with horizontal resolution up to a few kilometers. Comparisons between a simulation with resolved convection in the inner domain and a simulation using the Grell-Freitas convective parametrization showed similar performance in capturing transport (not shown). Thus, we used the Grell-Freitas convective scheme for both domains in all simulations to reduce the computation costs.**

[Figure]

*Figure 1 Average diurnal cycle of meteorology variables for observation at BAO 100m, simulation with resolved convection and simulation with Grell-Freitas convective parametrization. Averages are calculated for August 1 to 15, 2014.*

[Figure]

*Figure 2 measured and modeled (Grell-Freitas convective parametrization and resolved convection) ethane along C130 and P3-BAO and P3-PAO flights averaged for August flights.*

R1:

*Many informations on the chemical initial and boundary conditions (IC/BC) is missing: In which time interval are new BC data applied? Are the IC/BC data provided for this specific time interval/date? If yes, how do they fit to the two different dynamical boundary conditions used?*

Response:

Thank you. We have added more details regarding chemical IC/BC to the text.

Text:

Chemical boundary conditions from Monitoring Atmospheric Composition and Climate reanalysis (MACC**), available every 3 hours**,  (Inness et al., 2013) and model output from RAQMS**, available every 6 hours,** (Natarajan et al., 2012; Pierce et al., 2007) were used as chemical boundary and initial conditions **in the simulations. The model outputs from these global models are specific to the simulation time (24 July 2014 and end on 18 Aug 2014) and are interpolated to the WRF-Chem domain and temporal resolution prior to starting the simulations.**

*R1:*

*How many species are included in the used chemical mechanism?*

Response:

We have added more information on the hydrocarbon lumped groups. Because of complexity of chemical mechanism, we ask the readers to refer to Stockwell (1997) for details on species and reactions in the RACM chemistry mechanism.

Text:

We selected the Regional Atmospheric Chemistry Mechanism chemistry using Earth System Research Laboratory (RACM-ESRL) (Stockwell, Kirchner, Kuhn, & Seefeld, 1997) coupled to the Modal Aerosol Dynamics Model/Secondary Organic Aerosol Model (MADE/SORGAM). **RACM includes 17 stable inorganics, 4 inorganic intermediates, 32 stable organic species, and 24 organic intermediates**. RACM_ESRL (Kim et al., 2009) is an updated version of the RACM mechanism and includes 23 photolysis and 221 chemical reactions (Ahmadov et al., 2015). **To reduce the computational costs, hydrocarbons with similar behavior are lumped together. For example, "HC3" in the RACM_ESRL mechanism includes alkanes such as propane, n-butane, isobutane, and acetylene (ethyne), and alcohols such as methanol and ethanol and "TOL" includes toluene and benzene. Ethane and methane are treated exclusively in the RACM_ESRL mechanism. More details regarding the reactions and lumping groups can be found in Stockwell et al., 1997.**

Sect. 3:

*R1:*

*The title of the section is wrong. So far it provides only a listing of differences between the sensitivity simulations (results), but almost always neither an explanation for the differences is provided, nor is discussed, what the consequences for future simulations are (discussion).*

Response:

We have added more discussions in this section and replaced some of figures to provide better explanations for each sensitivity test.

R1:

*Table SM4 (maybe SM3) should become part of the main paper.*

Response:

Thank you for the comment. However, we decided to only point out the important differences between 10m and 100m and 300m in the text and keep10m and 300m figures in the SM. This is mainly due to similarities between these three figures.

R1:

*Figure 2 might provide a nice overview over all simulations, but it does not provide enough information for the comparison of the respective sensitivity simlations. Simply because of the number of lines (different symbols) not all simulations are identifyable. Here the supplement could be used to pro- vide a figure 2 like figure individually for each of the tested parametrisations (one for the PBL schemes, one for (re-)init, one for IC/BC, one for horizontal resolutions and on for the different emission scenarios).*

Response:

Thank you. We have added figures with separated sub-figures for each test in the supplement (Figure SM 1 to 3)

R1:

*Figures SM1 / SM2: some as for Fig. 2 add individual panels for those comparisons referring to SM2 (PBL and re-init). Maybe add reference to SM 2 also for the other tested parametrisations?*

Response:

Thank you. We have added references. We have added figures with separated sub-figures for each test in the supplement (Figure SM 1 to 3)

R1:

*Figure 3 and Figure 4: Same as for Fig. 2. Use the supplement to provide the plots individually for each sensitivity study, where the lines are hardly distinguishable otherwise.*

Response:

We have changed Figures 3 and 4 to whisker plots to be more distinguishable.

R1:

*Figure 3/4: dashed and solid lines are not distinguishable in the legend.*

Response:

We have added figures with separated sub-figures for each test in the supplement (Figure SM 1 to 3)

R1:

*Why are the tables for the model performance only provided for BAO station? It would be nice to be able to compare the so far provided statistics to the other stations listed in the introduction.*

Response:

We focused mostly on BAO because we had measurements not only at the surface but also at two higher altitudes (100m and 300m). However, we have used wind measurements from stations across the domain in other figures (Figure 7 and Figure SM 4).

R1:

*Tab. (4/SM3/SM4)(3/SM2)Why do you show observations in a row? In this way you have to repeat the value for the observations for each simulation. Just show obs in an additional column! In this way you could easily combine Tables 4, SM 3 and SM 4 and Tables 3 and SM 2 in one landscape table, respectively.*

Response:

Thank you. We have added a column with the observations in Tables 3 and 4 and Tables SM 1, 2, 3, and 4. We still believe that adding BAO 10 and 300m to the tables and figures in the main text is not necessary.

R1:

*p. 7, 26./27.: Here I disagree, not "all" simulations show better skill at daytime. Better change "all simulations" to "most simulations".*

Response:

Thank you. We changed the text "all simulations" to "most simulations".

Text:

Overall, **most** simulations show skill in capturing diurnal cycles of wind speed and direction with better agreement with observations for daytime

R1:

*p.8, l. 15 / l. 29: I am missing any information why the AM and PM flights are different and any suggestions, why the modeled concentrations are off by such a hugh amount. If these points would be sufficiently discussed in the following subsections, it would be ok to refer to the following analysis, but so far, they are not discussed in the further analysis either.*

Response:

As mentioned in the paper the differences in AM and PM measured and modeled ethane is because of lower PBL height in the AM flight. We have added more details regarding the differences in subsection 3.1 and in each sensitivity test subsection.

Sect. 3.1
R1:

*as already mentioned, the difference between the PBL simulations hardly visible in Fig. 2. It would be good to have a figure showing only the three simulations and the observations in the supplement.*

Response:

Thank you. We added figures separated by the sensitivity test to the supplement (Figure SM 1-3)

R1:

*Why do you use different chemical initial and boundary conditions for the PBL sensitivity studies (MACC) in contrast to all other sensitivity studies (Table 2)? This courses additional differences you never discuss.*

Response:

Thank you. We highlighted in the text that we only saw negligible differences in ethane concentrations when comparing MACC and RAQMS boundary conditions. Simulations using MACC and RAQMS IC/BC predicted similar ethane values thus we did not include it as one of the sensitivity tests in the discussion. Figure 3 illustrates the measured and modeled (MACC and RAQMS IC/BC) ethane concentrations along C130, P3-BAO, and P3-PAO August flights.

[Figure]

*Figure 3 measured and modeled (MACC and RAQMS Initial and Boundary Conditions) ethane along C130 and P3-BAO and P3-PAO flights averaged for August flights.*

Text:

**… Ethane concentrations showed no strong sensitivity to the two different chemical initial and boundary conditions (i.e., RAQMS and MACC) and is not discussed further.**

R1:

*p.9, l.16: Cite Figure SM1.*

Response:

Thank you. We cited the figure.

R1:

*Fig. 6: Article Text states "Fort-Collins-West (FCW)" , figure shows "FT".*

Response:

Noted

Thank you. We used "FC" throughout the paper

R1:

*p.9, l.26: For a non-specialist for boundary height measurements: Why is a quantitative comparison not possible?*

Response:

We removed the argument from the paper and only stated the different algorithm was used to estimated PBLH in the measurement.

Text:
Observed PBLH at the PAO, Fort Collins (FC), and Golden-NREL sites were retrieved from micro-pulse Lidar backscatter profiles **using Covariance Wavelet Transform (CWT)** (Compton et al., 2013).

R1:
*end of section: What are the consequences? Does the choice of PBL scheme matter or not? Which PBL scheme will you pick in future?*

Response:
We provide details on the impact of PBL scheme on NMB of ethane concentrations and the configurations we used for emission analysis in the conclusion section of the revised manuscript. As stated in the cover letter the main goal of this paper is not to find an optimal WRF-Chem configuration but to quantify the sensitivity of modeled fields to different model configurations with significant implications for e.g. inverse modeling studies.

Sect 3.2
R1:
*You change here from simulations driven by NCEP and MACC data, to simulations driven by ERAinterm and RAQMS data. Some information, why not the same driving data is used is missing here.*

Response:
Thank you for your comment. The simulations discussed in this section are based on the same meteorological and chemical IC/BC. The performance of NCEP and ERA-interim meteorological IC/BC is discussed in the section 3.4 Sensitivity to meteorological initial and boundary condition. As discussed in Figure 3 in this reply, we did not see a significant impact of chemical IC/BC on ethane fields. We added a sentence referring to Table 2 for details on model set-ups.

Text:
**Physical configurations and meteorological and chemical initial and boundary conditions are kept the same for these two simulations (Table 2).**

R1:
*p.10, l.28.: "in addition, Init 4 predicted higher wind speeds compared to Init5." And compared to Observations?*

Response:
Thank you. We changed the sentence.

Text:
In addition, Init4 predicted higher wind speeds compared to **BAO measurements (Figure 2) and** Init5.

R1:
*p.10, l.28.: Any suggestion, which simulation is more realistic?*

Response:
We provide details on the configurations we used for emission analysis in the conclusion. As stated in the cover letter the main goal of this paper is not to find an optimal WRF-Chem configuration but to quantify the sensitivity of modeled fields to different model configurations with significant implications for e.g. inverse modeling studies.

R1:
*p.10, l. 30/31 and 34 : Why does Init4 show the by far lowest ethane concentrations for C130-AM and all P3 flights? I thought the chemistry is not re-initialised in the same way as the meteorology?*

Response:
The difference is most likely due to higher wind speed in Init4. We added figure 8 to show the higher wind speed in Init4 across the domain.

Text:
When compared to C130-AM ethane concentrations (Figure 4), Init4 predicted the lowest ethane concentrations (a bias of -3.3 ppb and NMB of -63%) among all the simulations. **This is likely due to the high bias in wind speed in this simulation which resulted in lower concentrations of ethane (Figure 8).**

Sect. 3.3:
R1:
*The evaluation how much the simulation changes with respect to the meteorological initial and boundary data is meaningless without a comparison of the differences in the driving data themselves. How close are the reanalyses of ERA-Interim and NCEP-NFL to the observations in the area of interest?*

Response:

Thank you. We have compared WRF intermediate files prepared with ERA-Interim and NCEP-FNL model outputs in figure SM 6 and added discussion to the text.

Text:

**To prepare meteorological initial and boundary conditions from global models, WRF interpolates these outputs to the designed domains. Figure SM5 illustrates the differences between ERA-interim and NCEP-FNL model outputs interpolated to the outer domain at the lowest model level and averaged over August 1 to 15, 2014. Overall, the wind speed predictions by these two global models are very similar with slightly (less than 1 m/s) higher predictions by NCEP-FNL. ERA-interim and NCEP-FNL had larger discrepancies in temperature and relative humidity throughout the domain. Comparison with BAO observations (not shown) indicates similar performance for both models with somewhat lower temperature and higher relative humidity in ERA-interim compared to NCEP-FNL. These discrepancies did not have a large impact on temperature and relative humidity in the WRF-Chem simulation, however.** Figure 2 and Figure SM1-3 indicate that the performance of the two WRF-Chem simulations is comparable in capturing temperature and relative humidity **and better agreement with measurements during the day. This is because WRF-Chem only uses the global values as the initial and boundary values and resolves for atmospheric variables such as temperature and relative humidity in high resolution based on physical parametrizations set for the simulation**

R1:
*p. 11, l. 7-9: as far as I can deduce the correct lines from Fig. 2 and SM 1, both simulations overestimate temperature and underestimate moisture, where the ERA-Interim simulations is more off than the NCEP simulation.*

Response:
We modified the sentence for clarification to:

Text:

**Figure 2 and Figure SM1-3 indicate that the performance of the two simulations is comparable in capturing temperature and relative humidity with a better agreement with measurements during the day. Met5 had slightly higher temperature and lower relative humidity compared to Met6 and compared better to measurements especially during the night.**

R1:
*p.11, l. 14/15: I do not understand, I thought the Denver cyclone was not part of the analysis (top page 7). If this is just another "feature" and not the explanation of the huge differences in Met6*

*simulation for P3-PAO add this explanation and indicate by a line break, that you are talking about some- thing new.*

Response:
We removed the Denver cyclone discussion from the text to not deviate from the main goal and focus. Our analysis is focused on the time period Aug 1-15, 2014 and there were no occurrences of a Denver Cyclone during these two weeks.

R1:
*p. 11, l.19/20: Why do the simulations miss the event?*

Response:
We removed the Denver cyclone discussion from the text.

R1:
*p. 11, l.22 / SM4: Please show the absolute values (at least of one of the simulations) as well, otherwise absolute differences do not provide the full picture. This Figure should be part of the main paper. If you want to discuss the Denver cyclone, figure SM4 should be part of the main paper. But in this case, the analyses should go beyond the mere stating of the differences.*

Response:
Thank you. We removed the Denver cyclone discussion from the text

Sect 3.4:
R1:
*p. 11, l. 27: No, the domains are not run "independently" as the 4x4 km2 domain depends on the 12x12 km2 domain.*

Response:
We added details to clarify.

Text:
**This means that while the outer domain provides the boundary conditions to the inner domain, the higher resolution fields from the inner domain do not alter the outer domain fields.**

R1:
*p.11, l. 28./29.: What about temperature and relative humidity at PAO and WC tower?*

Response:

We have added more information on model performance in capturing wind in other stations across the domain because of its larger impact on ethane.

R1:

*p. 11, l. 28 /31: The different labels are hard to detect in the Figure 2 / SM2.*

Response:

Thank you. We added figures separated by the sensitivity test to the supplement (Figure SM1-3).

R1:

*p. 11, l. 31. This is only true at nighttime (Tab. 4 / SM 4). Cite the tables in the text.*

Response:

Thank you. We changed the sentence.

Text:

At 100m and 300m altitudes at BAO, the coarse domain predicted higher **nighttime** wind speed compared to the fine domain **and the measurements**.

Sect. 3.5:

R1:

*p. 12, l. 16: According to caption, Fig. 8 shows measurements. How can it be seen, that the model does not match the measurements? This is seen from Fig. SM 5, therefore this figure belongs in the article, not in the supplement. The phrase "as shown in Fig. 8a" is very irritating. It belongs in front of the "which", i.e., after the statement about high values of ethane.*

Response:

Thank you. We merged Figure 8 and Figure SM 5 and changed the sentence.

Text:

**Figure 10a and b illustrate** high values of ethane concentrations in the vicinity of oil and NG facilities which were not captured by the model resulting in low biases.

R1:

*as the comparison of the two emission scenarios is the topic of this subsection, SM 6 also belongs into the main paper. However, Fig. 9 could be part of the Supplement.*

Response:

Thank you for your comment. We agree that Figure SM 8 includes important information regarding the sensitivity of ethane to oil and gas emission. However, we believe that Figure 9 can provide more information about the background values and magnitude of underestimation. To keep the paper short, we had to put this figure in the supplement material.

R1:

*p.13, l.18/19: I do not think that you need an indicator for low emission rates. These are input by the emission inventory, thus this is aprior knowledge.?*

Response:

Thank you. Yes, emission rates are fed into the model through the emission inventory and the emission inventory is amongst the largest sources of uncertainties in the model. Here, we find a pronounced low model bias for xylene and toluene concentrations in the vicinity of oil and gas facilities. The model performance was not improved in the simulation with doubled oil and gas emission. This likely due to lower emission rates for these species compared to the measurements.

R1:

*Discussion of Fig. 10: Say more about the best fit lines (meaning of differences in intercepts and slope).*

Response:

We added more information about the differences between intercepts and slops.

Text:

Figure 10 illustrates the HC3 to TOL ratio measured along the C130 PM **limited to NFR region and altitudes below 2000m** and the corresponding model values. Figure 10a shows oil and NG influenced points with enhanced measured ethane (concentrations greater than 2 ppb). HC3 to TOL ratios in oil and NG influenced locations show inconsistency between measured **(HC3/TOL = 68)** and Em7 modeled ratios **(HC3/TOL = 22)** which was improved in the Em8 **(HC3/TOL = 40.9)**. However, doubling oil and NG emission still resulted in underestimations of HC3, TOL, and their ratios in this region. Figure 10b shows urban influenced points with low measured ethane (concentrations less than 2 ppb). Modeled HC3 to TOL **ratios (7.3 for Em7 and 8.9 for Em8)** in the urban influenced locations did not show large sensitivity to oil and NG emissions and agreed well with the measurements **(10.2). In both oil and NG and urban influenced regions models predicted lower than measured Y-intercepts which was not improved in Em8. Figure9c also confirms the low bias (about -2ppb) in background HC3 in the model. One reason for this offset can be underestimation in HC3 concentration in the boundary condition fields or leakage from the NG distribution system which was not captured in the model.**

R1:

*p. 13, l. 31/32: Where do I see these low biases (cite respective table / figure)*

Response:
Thank you. We have added reference to figure 9.

Text:
The low model bias for these species is more pronounced compared to the low model bias in ethane **(Figure 11).**

R1:

*What about cross evaluation? What changes with different chemical IC/BC? What is the influence of the PBL scheme for the higher emission scenario?*

Response:
Modeled ethane concentrations were not sensitive to the chemical IC/BC (Figure 3). Running all the sensitivity simulations with doubled oil and gas emissions is computationally very expensive and not much can be gained from these simulations. A simulation with doubled oil and gas emissions was conducted to assess the sensitivity of ethane to oil and gas emissions and will be later used to perform a variational inversion algorithm with the goal to find an optimal scaling factor for oil and gas emissions. If significant improvement is achieved with this method, we will use simulations discussed in this work to calculate the confidence level around optimal scaling factor.

Conclusion
R1:

*p. 14, l.10-22: In this section "NMB" and "NMB variability" is falsely used synonymously. It might be true that the inter-simulation variability of NMB might be usable "as a proxy for variability in the model performance caused by model configuration". But the percentages listed below are only the NMBs and not the inter-simulation variability of the NMB.*
*However, this paragraph again only lists (new) results (not even a discussion). Therefore it does not belong in the conclusion section.*

Response:
Thank you. We made sure that we did not falsely use NMB and NMB variability synonymously. We added NMB variability for each sensitivity test and each flight to Table 5. We added a concluding sentence on which configuration we used for further emission sensitivity analysis. We also made sure not to include any new results in the conclusion section.

Text:

**To further investigate the performance of the model in capturing oil and NG emissions in the NFR we used a similar domain set-up with 12 km x 12 km and 4 km x 4 km horizontal resolution for outer and inner domains, respectively, daily re-initialization of meteorological variables with ERA-interim model, and MYNN3 PBL scheme.**

R1:

*p. 14, l. 24 - p.15, l. 4: This section stays very in-concrete. All the species discussed here have a medium range lifetime. Nevertheless, the aspect of the influence of chemical initial and boundary conditions is completely left out.*

Response:
We added details regarding this issue in section 3.6.

Text:
Figure 10 illustrates the HC3 to TOL ratio measured along the C130 PM **limited to NFR region and altitudes below 2000m** and the corresponding model values. Figure 10a shows oil and NG influenced points with enhanced measured ethane (concentrations greater than 2 ppb). HC3 to TOL ratios in oil and NG influenced locations show inconsistency between measured **(HC3/TOL = 68)** and Em7 modeled ratios **(HC3/TOL = 22)** which was improved in the Em8 **(HC3/TOL = 40.9)**. However, doubling oil and NG emission still resulted in underestimations of HC3, TOL, and their ratios in this region. Figure 10b shows urban influenced points with low measured ethane (concentrations less than 2 ppb). Modeled HC3 to TOL **ratios (7.3 for Em7 and 8.9 for Em8)** in the urban influenced locations did not show large sensitivity to oil and NG emissions and agreed well with the measurements **(10.2). In both oil and NG and urban influenced regions models predicted lower than measured Y-intercepts which was not improved in Em8. Figure9c also confirms the low bias (about -2ppb) in background HC3 in the model. One reason for this offset can be underestimation in HC3 concentration in the boundary condition fields or leakage from the NG distribution system which was not captured in the model.**

R1:

*For me the final conclusion is missing: Based on this study, what has to be taken into account by setting up a WRF-Chem simulation? What would be the best WRF-Chem setup?*

Response:
We have added details regarding the WRF-Chem configurations we used for emissions sensitivity analysis. As stated in the cover letter the main goal of this paper is not to find an optimal WRF-Chem configuration but to quantify the sensitivity of modeled fields to different model configurations with significant implications for e.g. inverse modeling studies. Overall, changing one WRF-Chem model configuration may improve the performance of the model in

some aspect of atmospheric chemistry and transport and may adversely impact another aspect. Thus, the "best configuration" can depend on the goal of the study.

Minor points:

R1:

*From my point of view WRF-Chem is not a Chemical Transport Model (CTM). The term "transport" includes the fact, that the chemical substances are transported, but they do not feed back to the dynamics. WRF-Chem is a fully featured regional atmospheric chemistry model (RACM) which very well includes e.g., radiation and cloud feedback. Therefore I recommend to omit the term CTM with respect to WRF-Chem. I know that whole communities use this term wrongly, but this did not change the misconception of it.*

Response:

We removed the term CTM.

Text:

High resolution three-dimensional **atmospheric chemical transport models** can better capture the variability in meteorology and chemistry in different domains. **Paired with observations, these models** help evaluate the performance of emission inventories on high temporal and spatial scales and allow assessments of the impact of oil and NG activities on regional air quality.

R1:

*As this article is very WRF-Chem specific, it would be apropriate to name the model in the title of the article, e.g. "Impacts of physical parametrizations on pre- diction of ethane concentrations for oil and gas emissions a case study with WRF- Chem v3.6.1" or 'Impacts of physical parametrizations of WRF-Chem (v3.6.1) on prediction of ethane concentrations for oil and gas emissions"*

Response:

We changed the title.

Text:

Impacts of physical parametrization on prediction of ethane concentrations for oil and gas emissions **in WRF-Chem**

R1:

*Sect. 2.3: Add more information on how the emissions of 76 species are mapped to the species of the chemical mechanism used.*

Response:

Thank you. We have added Table SM 5 to the supplemental material. This table is copied from the subroutine (Fortran code) provided by NOAA to convert species in the NEI-2011 emission inventory to RACM and MADE/SORGAM chemical and aerosol mechanism.

Text:

**Table SM 5 includes details on the mapping table used to convert NEI-2011 species to RACM and MADE/SORGAM chemical and aerosol mechanism.**

R1:

*p.7, ll. 5-17: add the information, that these results will be discussed in more detail in the following subsections.*

Response:

Thank you. We have added subsection 3.1. Evaluation of overall model performance to better address this issue.

R1:

*p.7, ll. 19-34: add the information, that these results will be discussed in more detail in the following subsections.*

Response:

Thank you. We have added subsection 3.1. Evaluation of overall model performance to better address this issue.

R1:

*p. 10, l. 10: cite Table 5*

Response:

Thank you. We cited Table 5.

Text:

**Table 5 summarizes the mean and NMB for all simulations using C130 and P3 ethane measurements.**

R1:

*p. 10, l. 11: cite Fig. 4.*

Response:

We cited Figure 4.

Text:

**Similar to the C-130 comparison, Figure 4, the simulations did not capture the high ethane values measured during P3-BAO and P3-PAO spirals.**

R1:

*S1: Sums over i without any "i"s in the equations? What is "n"? IOA is introduced (also in the article) but never referenced.*

Response:

Thank you for your comment. We fixed the problem in equations. We removed IOA from tables.

Typos & CO
R1:
*p.4, l. 26: remove "Tables and Figures"*
Response:
We removed "Tables and Figures"

*p.5, l. 8: Did you consciously choose an abbreviation (4-MnERi) which is not included in Table 2 ?*
Response:
Thank you for your comment. No, this was a typo. We changed the example to 5-MnERi

*• p.5, l. 9: The abbreviation of the PBL scheme is (MYNN3)*
Response:
Thank you. We fixed the typo.

*• Through out the article (and in the figures) inconsistent abbreviations for Fort- Collings West are used (FC, FCW ...). This should be uniform.*
Response:
Thank you. We used FC through the text.

*• Fig. 5: Whatisthemeaningofthelineat41N?*
Response:
That is a state line between Colorado and Wyoming.

*It's the state line*
*• p. 9. l. 11: Cite SM 3/4 at end of sentence.*
Response:
Thank you. We cited the figures.

*• p. 9. l. 11: Fig. 5 does not "compare" anything, it "visualises", "shows", "depicts" ...*

Response:

We changed "compares" to "shows"

*• p. 9, l. 30 / 31: use full sentences: "PBL1, non-local PBL scheme" → " PBL1 using a non-local PBL scheme".*

Response:

We updated the sentence:

Text:

**The local PBL schemes (i.e.** PBL2 and PBL3) predict cooler and moister climates and lower PBLH, which indicates less vertical mixing.

*• p. 11. l. 20: A figure does not "compare" anything, it "visualises", "shows", "depicts" ...*

Response:

We changed "compares" to "shows"

*• p. 12, l. 14, Fig. 8: Limited to 200agl or 1500 m ?*

Response:

Thank you for the comment. We updated the sentence.

Text:

… the C130 PM measurements and bias limited to altitudes below **2000m.**

*• Fig. 8.: If these are measurements, what does "grids with more than 4 measurement points" mean?*

Response:

Thank you. We updated the caption.

Text:

Figure 8. Mean and mean bias for ethane concentrations (a and b), CO (c and d), HC3 (e and f), and TOL (g and h) along the C130 PM flights are limited to measurements below 2000m agl **and grids with more than 4 measurement points**. The outline of Denver county and the locations of BAO and PAO are marked on the underlying terrain map.

*• Fig. 10: Improve caption. The second sentence sound as if model vs. measure- ments are plotted.*

Response:

Thank you. We updated the caption.

Text:

Figure 10. Scatter plot of HC3 vs. TOL concentrations along the C130 PM flights limited to measurements in the NFR and below 2000m altitude. Plot (a**) shows HC3 vs. TOL (when measured ethane is greater than 2ppb) for measurements and the corresponding model values. Plot (b) shows HC3 vs. TOL (when measured ethane is less than 2ppb) for measurements and the corresponding model values**. Grey circles represent measurements, red diamonds represent the Em7 (base emissions), and blue circles represent Em8 (perturbed emissions). Grey, Red and Blue lines show the best fit using least square linear regression method for observations, Em7 and Em8, respectively.

*• Fig. 10 : give parameters of best fit line.*
Response:
We added parameters to the figure

*• Tab. 5: Is the NMB for PBL3 P3-BAO really 0 ?*
Response:
Yes, with 1 decimal point of ethane (ppb).

*• SM2: caption: state that "surface winds" are shown.*
Response:
We added information regarding the height of the measurement.

*• Table SM4: omit line breaks for init5*
Response:
We updated the Table.

*• Fig. SM3: What is surface ethane: ethane in the lowest model layer?*
Response:
Yes, surface ethane concentrations are represented by ethane fields at the lowest model layer which ranges from about 0 to 20m altitude agl.

**Reviewer 2**

Thank you very much for your helpful comments and suggestions. Please find replies to individual comments below.

Major Comments

*R2:*

*Aim, approach and conclusions: The aims of the presented study are somewhat twofold. 1) evaluate the chemistry-transport model and 2) draw some preliminary conclusions on the emissions from oil and gas extraction in the Northern Front Range Metropolitan area (NFRMA). In general, I agree with the authors that a transport model needs to be validated before it can be used for inverse modelling purposes (here emission estimation). However, these two subjects are somewhat mixed together in the study, since one main evaluation parameter of the model system is the ethane concentration, which strongly depends on the emissions, which one finally would like to determine. The manuscript would gain in significance if these two subjects (transport model performance and emissions from oil and gas) could be separated more clearly in the presentation of the results. Instead of evaluating model performance for ethane another model tracer (e.g. carbon monoxide) should be given more attention during the evaluation part of the manuscript. Since carbon monoxide is largely unrelated to the oil and gas emissions and its emissions are otherwise relatively well know, one should be able to distinguish, which model configuration is best suited for tracer transport simulations. Only afterwards, the less well known ethane emissions should be discussed.*

Response:
We appreciate the comment on using another tracer to separate the discussion on transport from oil and gas emissions. However, we found ethane to be the best tracer to study the transport of oil and gas emissions in the region. Besides the complex transport patterns in the region, there are multiple emission sources which makes finding an effective tracer very complicated. For example, CO can be released from any combustion sources such as cars, power plants, and oil and gas extraction activities. Also, it is sensitive to the background value. As discussed in the paper, high concentrations of CO were measured downwind of Denver as of well as oil and gas facilities. Comparison with modeled CO showed overestimation of CO over Denver and underestimation over oil and gas facilities. The oil and gas sector is the only notable source for ethane in the NFR which can help us attribute the errors in emissions to this sector. Ethane concentrations in the NFR also do not show any notable influence from transport from outside the NFR.

*R2:*

*In this context another question arises concerning the application of the full chemistry version of WRF-CHEM in the sensitivity runs. A passive tracer setup (as available in WRF-CHEM) would have been sufficient to carry out the sensitivity runs as well. Only the discussion including shorter lived VOCs really grants running the complete model. This is more a comment for future sensitivity simulations than for the current manuscript. However, since the complete chemistry model was run, it would also be interesting to have a look at a secondary pollutant (e.g., ozone) and how it reacts to the emission variations in the final sensitivity experiments. Most likely ozone observations were available from all flight campaigns and additionally surface sites. The model-observation mismatch for ozone might deliver another hint towards the suspected underestimation of emissions from gas and oil extraction.*

Response:
The FRAPPÉ and DISCOVER-AQ dataset includes comprehensive ground-based and airborne measurements of ozone and ozone precursors. Discussion on ozone predictions and ozone production in the model is complicated by uncertainties in the anthropogenic and biogenic emissions estimates and the complexity of the transport in the region. We decided to focus in this paper on the meteorology and the/transport and magnitude of primary oil and gas emissions as trying to understand all uncertainties would be too large a topic for a single manuscript.

*R2:*

*Finally and coming back to the beginning, the conclusions of the paper should give more emphasise to the finally selected model configuration. Which is the best configuration, which parameters were most important to change, and why does the change make sense? Without such a recommendation (although not necessarily valid for other study areas or periods) the whole presentation of sensitivity results remains of no avail for the reader who wants to find concrete hints for setting up his/her own simulations. Right now the conclusion only summarises all sensitivity simulations together, although there were clear performance difference for some of the runs (PBL1 vs PBL3, free-run vs re-ini, etc). These are important details of the model setup that should be more easily accessible when browsing the paper.*

Response:
We have added conclusions on the model configurations we used for the emissions sensitivity analysis and highlighted the parameters we found most important. As we stated in the cover letter: The main goal of this paper is not to find an optimal WRF-Chem configuration but to quantify the sensitivity of modeled fields to different model configurations with significant implications for e.g. inverse modeling studies. We clarified this in the revised manuscript.

Overall, changing one WRF-Chem model configuration may improve the performance of the model in some aspect of atmospheric chemistry and transport and may adversely impact another aspect. Thus, the "best configuration" can depend on the goal of the study.

R2:
*Page 2, Line 9: From the context and my own knowledge of emissions from the oil and gas sector it remains unclear why NOx emissions should play a major role in the extraction process. Is it due to fossil-fuel operated machinery (compressors, pumps, etc) or due to flaring? In the discussion of the model results there is no more mentioning of NOx either. Was it not observed on-board the aircraft?*

Response:
Thank you for your comment. We have added details about the NOx emissions from the oil and gas sector. NOx was measured at ground level and on-board the aircraft during FRAPPÉ and DISCOVER-AQ. Because of the complexity in estimating NOx emissions due to its short life-time and having multiple sources other than the oil and gas sector we did not include it in this paper. However, we have included multiple references that studied NOx and ozone concentrations in the Northern Front Range during FRAPPÉ and DISCOVER-AQ.

Text:
With the rapid increase in the unconventional oil and NG production, higher than expected levels of greenhouse gases, specifically methane, and air pollutants such Non-Methane Hydrocarbons (NMHC) and NOx **(from compressors, pneumatic devices, trucks, and other equipment using fossil fuel (Allen, 2016; Olaguer, 2012))** have been observed in some places in vicinity of oil and NG facilities.

R2:
*P3, first paragraph: Others have used WRF-CHEM already for 'inverse modelling' of emissions from oil and gas extraction. The study by Barkley et al. (2017), which used WRF-CHEM for CH4 emission estimation (although not with full chemistry), should be mentioned as well. Either when mass balance approaches to estimate emissions are discussed or more general in the discussion of the results.*

Response:
Thank you. We added the reference on introduction.

Text:
**Paired with observations and using inverse modeling techniques, these models** help evaluate the performance of emission inventories on high temporal and spatial scales **(Barkley et al.,**

**2017; Cui et al., 2014, 2017)** and allow assessments of the impact of oil and NG activities on regional air quality.

R2:

*Subsection 2.1: The presentation of the observations is lacking any reference to the applied measurement techniques, data quality, etc. Some of this information is given later when the results are presented, but it should actually be presented in this subsection. For the more involved measurements (VOC concentrations) it would also be nice to learn a bit about observation uncertainties before comparing with model results.*

Response:

Thank you for your comment. We moved the information about the ethane measurements to subsection 2.1.

R2:

*P4,L30: The kind of VOC lumping used in RACM should be mentioned. Is ethane treated explicitly? How many chemical species are present?*

Response:

Thank you we added details on the VOC lumping to the text.

Text:

**To reduce the computational costs hydrocarbons with similar behavior are lumped together. For example, "HC3" in the RACM_ESRL mechanism includes alkanes such as propane, n-butane, isobutane, and acetylene (ethyne), and alcohols such as methanol and ethanol. "TOL" includes toluene and benzene. Ethane and methane are treated exclusively in the RACM_ESRL mechanism. More details regarding the reactions and lumping groups can be found in Stockwell et al., 1997."**

R2:

*P5,L21 and elsewhere: Since it was mentioned above (P5,L10) that the simulation ID according to table 2 would be used throughout the text to identify the sensitivity simulations, I see no need to give the simulation name here as well. All these abbreviations will only confuse. All figures also contain a mix of names and identifiers that are not according to the ID. This should be unified, so that the reader only has to follow one set of abbreviations.*

Response:

Thank you. We unified the IDs used in figures and in the text. Some of the simulations such as 5-MnERi were used in multiple sensitivity tests. To avoid confusion, we used IDs in all figures.

R2:

*P6, 1st paragraph: What is the rational for this re-initialisation? The model domain is relatively small. Shouldn't the BCs dominate anyway? In this context it should also be mentioned why no meteorological data assimilation was run for WRF, which probably would have reduced the spread in the sensitivity tests significantly.*

Response:

Thank you for your comment. The outer domain covers most part of the CONUS with 12 km x 12 km resolution and receives BC from the global re-analysis. The outer domain provides BCs for the inner domain (covering Colorado and Utah) with the inner domain not having any feedback on the outer domain. Thus, we believe that BC do not dominate at least in the inner domain. Data assimilation and measurements are used to produce global models used for re-initialization of the model. By re-initializing meteorological field every 24 hours using re-analysis products from global models we decided not to use any nudging (data assimilation) in our simulations. One can test and compare the model performance with different nudging systems vs. re-initialization. However, given the complex terrain in the NFR and the limited amount of measurements nudging may not necessarily add strong constrains.

R2:

*P7, L5: 'average error'. Should this be absolute error, which would actually fit the definition in the supplement.*

Response:

Thank you. We changed "averaged" to "absolute" in the text and in the supplemental material.

Text:

For quantitative comparison between the simulations, we used statistical measures including correlation coefficient (R), root mean square error (RMSE), mean **absolute** error (MAE), mean bias (MB), normalized mean bias (NMB), and index of agreement (IOA).

R2:

*P7, L6: Although the supplement provides the definition of the comparison parameter 'index of agreement', it would be nice to give a short interpretation of its benefits and expected value range for a well performing simulation. Furthermore, values of IOA are presented in a table, but never discussed in the text. Something that is true for other comparison parameters as well. If they are shown in the table a short description and interpretation in the text should also be given.*

Response:

Thank you. We removed IOA from tables to focus on statistics that was used in the text.

R2:

*P7, L9: I feel like one could start a new subsection here (after 'model values') that deals with the general results. Everything before is an introduction to the results, now the first real results are shown.*

Response:

Thank you for your comment. We added a subsection (3.1. Evaluation of overall model performance) in this section.

R2:

*P7, L25: Maybe I missed this somewhere else, but at which altitude were the measurements at WC tower and PAO taken?*

Response:

Altitudes were added.

R2:

*P8, L12 and 8ables or figures? If yes a cross-reference to the figure would be useful, if not this should also be mentioned (e.g. adding 'not shown'). Otherwise the reader will try to confirm this information in one of the figures.*

Response:

Thank you. We cited Table 5 here.

Text:

Lower concentrations of ethane were measured during the PM flights compared to AM flights because of the higher PBLH and stronger vertical mixing in the afternoon. **Table 5 summarizes the mean and NMB for all simulations using C130 and P3 ethane measurements.** In all simulations, the ethane concentrations are under-predicted by up to 3.3 ppb (NMB ranges between -63% to -42%) for the C130 AM flights and up to 1.7 ppb (NMB ranges between -47.6% to -29.5%) for the C130 PM flights.

R2:

*P8, L31ff: It is argued that the model 'represents the profile shape [...] well.' I don't agree with this completely and for all cases. For the afternoon observations, where C130 and BAO observations actually show relatively low values at the bottom of the profile, most model runs still show a smoothly increasing concentration towards the surface. What is the potential reason for the observed profile shapes and what does the model miss here?*

Response:

Thank you. We have changed Figure 3 and 4 to whisker plots at each altitude bin to convey more statistical information. We have changed the sentence discussing the vertical profile shape.

Text:

**While the model shows difficulty in representing the absolute magnitude in ethane concentrations in all simulations at lower altitudes, most simulations capture the changes in variance of ethane concentrations from lower to higher altitudes well especially for the C130 and P3 BAO flights.**

R2:

*P9, L26: It is argued that a 'quantitative comparison between model and measurement [of PBLH] is not possible'. I don't agree. It depends on the situation, but during the day the agreement between a lidar-derived PBLH and one derived from temperature profiles should actually be pretty good as long as you have an aerosol-laden PBL, which is most likely the case in the American South-West. See also Collaud Coen et al. (2014).*

Response:

We removed the argument from the paper and only stated the different algorithm was used to estimated PBLH in the measurement.

Text:

Observed PBLH at the PAO, Fort Collins (FC), and Golden-NREL sites were retrieved from micro-pulse Lidar backscatter profiles **using Covariance Wavelet Transform (CWT)** (Compton et al., 2013).

R2:

*Figure6: I suggest to extend this figure for all other sensitivity simulations as well. Next to wind speed, PBLH is the most important dispersion parameter on local to regional scale. The use of the same turbulence scheme in all other sensitivity runs will not necessarily lead to the same PBL heights. For example the two different BC runs are also likely to result in different PBL structures.*

Response:

Thank you. In Figure 3 we plotted cross sections of modeled ethane for all sensitivity tests at PAO averaged from Aug 1 to 10, 2014 and measured PBL height by Lidar backscatter. This figure can help us compare the diurnal variation of measured PBL height and its impact on ethane vertical distribution.

R2:

*P10,L7ff: This paragraph should include a short description of the changes in profile shape for the different PBL runs. The interesting part is that with PBL2 and PBL3 and although these runs are supposed to produce more realistic PBL heights, the increase in PBL concentrations compared to PBL1 is opposite to what the observations suggest (Fig 3 and 4). That is: afternoon concentrations increased from PBL1 to PBL2 and PBLE3 below a certain altitude, but the resulting profile shape is then even less in line with the observations.*

Response:

Thank you for the comment. We replaced figures 3 and 5 with whisker plots at each altitude bin. Figure 3 shows a faster growth in morning PBL in PBL1 compared to PBL2 and PBL3. This resulted in higher concentrations of ethane in the morning to noon period at higher altitudes (0.5 to 2 km). We added this discussion in the paper.

Text:

**On average PBL1 predicted higher ethane concentrations during AM flights at lower altitudes compared to PBL2 and PBL3 (Figure 3). Faster evolution of morning PBL and stronger vertical mixing in PBL1 lofted pollutants (including ethane) higher into the atmosphere in the morning (Figure 6). The rapid growth of morning PBL in PBL1 resulted in higher concentration of ethane at higher altitudes (0.5 to 2 km) compared to PBL2 and PBL3.**

R2:

*P11,L14ff: In the beginning of the results section it is mentioned that the Denver cyclone period was removed from the analysis. Why is it presented here again? Either this part should be moved to the beginning of the results section when justifying why the phase was excluded from the analysis or it should be mentioned here again which period is really contained in the figures.*

Response:

We removed the Denver cyclone discussion from the text to not deviate from the main goal and focus. Our analysis is focused on the time period Aug 1-15, 2014 and there were no occurrences of a Denver Cyclone during these two weeks.

R2:

*Section 3.5: Megan biogenic emissions are mentioned in the Table but not in the text. What is their significance for this sensitivity run and overall for this study? Did Megan predict significant emissions of ethane or HC3? Probably not!*

Response:

Thank you. We added a sentence to the text.

Text:

We used the Model of Emissions of Gases and Aerosols from Nature (MEGAN) for biogenic emission in all simulations (Guenther et al., 2012). **Ethane does not have a significant biogenic source (Yacovitch et al., 2014); thus, we did not assess the impact of biogenic emissions in this study.**

R2:

*P12,L27f: Not surprisingly, the measurements did not differ for the two different simulations!*
*Should it be 'simulated CO'?*

Response:
Thank you. We corrected the sentence.

Text:
 CO is mostly emitted from combustion processes and is released from many different source
sectors. **Figure10.c shows CO enhancements over both Denver and oil and NG facilities.**

R2:
*P13,L21ff: In the discussion of Figure 10 it should be mentioned that for the longer-lived HC3*
*one should/could have removed a background concentration. The presentation is still valid if*
*only the slopes/ratios are discussed and background did not change over the domain of interest.*
*The latter should be commented on.*
*P13,L26f: Following up on the last comment, it most likely is not recently emitted HC3 that leads*
*to elevated levels but background levels. The alkanes contributing to HC3 are relatively long-*
*lived and the HC3 concentration at the domain boundaries is most likely a few ppb. This could*
*be easily checked in the model fields. In contrast TOL is relatively short-lived, so BC*
*concentration are probably close to zero.*

Response:
Thank you. We have added this to the discussion.

Text:
Figure 10 illustrates the HC3 to TOL ratio measured along the C130 PM **limited to NFR region**
**and altitudes below 2000m** and the corresponding model values. Figure 10a shows oil and NG
influenced points with enhanced measured ethane (concentrations greater than 2 ppb). HC3 to
TOL ratios in oil and NG influenced locations show inconsistency between measured
**(HC3/TOL = 68)** and Em7 modeled ratios **(HC3/TOL = 22)** which was improved in the Em8
**(HC3/TOL = 40.9)**. However, doubling oil and NG emission still resulted in underestimations of
HC3, TOL, and their ratios in this region. Figure 10b shows urban influenced points with low
measured ethane (concentrations less than 2 ppb). Modeled HC3 to TOL **ratios (7.3 for Em7**
**and 8.9 for Em8)** in the urban influenced locations did not show large sensitivity to oil and NG
emissions and agreed well with the measurements **(10.2). In both oil and NG and urban**
**influenced regions models predicted lower than measured Y-intercepts which was not**
**improved in Em8. Figure9c also confirms the low bias (about -2ppb) in background HC3 in**
**the model. One reason for this offset can be underestimation in HC3 concentration in the**

**boundary condition fields or leakage from the NG distribution system which was not captured in the model.**

R2:

*P2,L25: 'Thousand Cubic Feet' probably supposed to be 'Million Cubic Feet'. Also, I am not sure if ACP allows US customary units.*

Cubic meter was added to the text.  MCF is short for thousand cubic feet.

R2:

*P9,L22 and elsewhere: 'Fried, 2000' is not the correct formatting. Please refer to the ACP guidelines.*

Response:

Thank you. We corrected the format.

[revised manuscript text omitted]

$$RMSE = \sqrt{\frac{\sum_{i=1}^{n}\left(C_{p_i} - C_{o_i}\right)^2}{n}} \tag{2}$$

$$MAE = \frac{1}{n}\sum_{i=1}^{n}\left|C_{p_i} - C_{o_i}\right| \tag{3}$$

$$MB = \frac{1}{n}\sum_{i=1}^{n}\left(C_{p_i} - C_{o_i}\right) \tag{4}$$

$$NMB = \frac{\left(\overline{C_p} - \overline{C_o}\right)}{\overline{C_o}} \times 100\% \tag{5}$$

Where $C_o$ is the observation value, $C_p$ is the model value, $\sigma$ is the standard deviation, $\bar{C}$ is the mean value, and n is total number of observation points

Table SM 1 Conversion table used to map species from NEI-2011 emission inventory to RACM chemical mechanism in and MADE/SORGAM aerosol module

| Emission inventory name | WRF-Chem name | Weight | Species name |
|---|---|---|---|
| CO | e_co | 1.00 | Carbon monoxide |
| NOX | e_no | 1.00 | Nitrogen Oxides (NO or NO2) |
| SO2 | e_so2 | 1.00 | Sulfur dioxide |
| NH3 | e_nh3 | 1.00 | Ammonia |
| HC01 | e_ch4 | 1.00 | Methane |
| HC02 | e_eth | 1.00 | Ethane  kOH<500 /ppm/min |
| HC03 | e_hc3 | 1.00 | Alkane 500<kOH<2500 exclude(C3H8,C2H2,ethanol,acids) |
| HC04 | e_hc3 | 1.11 | Alkane 2500<kOH<5000 exlude(butanes) |
| HC05 | e_hc5 | 0.97 | Alkane 5000<kOH<10000 exlude(pentanes) |
| HC06 | e_hc8 | 1.00 | Alkane kOH>10000 exclude(ethylene glycol) |
| HC07 | e_ol2 | 1.00 | Ethylene |
| HC08 | e_olt | 1.00 | Alkene kOH <20000 /ppm/min |
| HC09 | e_oli | 1.00 | Alkene kOH >20000 /ppm/min exclude(dienes,styrenes) |
| HC10 | e_iso | 1.00 | Isoprene |
| HC12 | e_tol | 1.00 | Aromatic kOH <20000 /ppm/min exclude(benzene and toluene) |
| HC13 | e_xyl | 1.00 | Aromatic kOH >20000 /ppm/min exclude(xylenes) |
| HC14 | e_hcho | 1.00 | Formaldehyde |
| HC15 | e_ald | 1.00 | Acetaldehyde |
| HC16 | e_ald | 1.00 | Higher aldehydes |
| HC17 | e_ald | 1.00 | Benzaldehyde |
| HC18 | e_ket | 0.33 | Acetone |
| HC19 | e_ket | 1.61 | Methylethyl ketone |
| HC20 | e_ket | 1.61 | PRD2 SAPRAC species (aromatic ketones) |
| HC21 | e_hc3 | 0.40 | Methanol |
| HC22 | e_ald | 1.00 | Glyoxal |
| HC23 | e_ald | 1.00 | Methylglyoxal |
| HC24 | e_ald | 1.00 | Biacetyl |
| HC25 | e_csl | 1.00 | Phenols |
| HC26 | e_csl | 1.00 | Cresols |
| HC27 | e_ald | 0.50 | Methacrolein |
| HC27 | e_olt | 0.50 | Methacrolein |
| HC28 | e_ket | 0.50 | Methylvinyl ketone |

| | | | |
|---|---|---|---|
| HC28 | e_olt | 0.50 | Methylvinyl ketone |
| HC29 | e_ket | 1.00 | IPRD SAPRAC species (>C4 unsaturated aldehydes) |
| HC31 | e_ora2 | 1.00 | Acetic Acid |
| HC32 | e_ora2 | 1.00 | >C2 Acids  (SAPRC PACD species) |
| HC33 | e_csl | 1.00 | Xylenols  (SAPRC-11 species) |
| HC34 | e_csl | 1.00 | Catechols  (SAPRC-11 species) |
| HC36 | e_olt | 1.00 | Propylene |
| HC37 | e_hc3 | 0.40 | Acetylene |
| HC38 | e_tol | 0.29 | Benzene |
| HC39 | e_hc3 | 1.11 | Butanes |
| HC40 | e_hc5 | 0.97 | Pentanes |
| HC41 | e_tol | 1.00 | Toluene |
| HC42 | e_xyl | 1.00 | m-Xylene |
| HC43 | e_xyl | 1.00 | p-Xylene |
| HC44 | e_xyl | 1.00 | o-Xylene |
| HC45 | e_hc3 | 0.57 | Propane |
| HC46 | e_oli | 1.00 | Dienes |
| HC47 | e_olt | 1.00 | Styrenes |
| HC47 | e_tol | 1.00 | Styrenes |
| HC48 | e_hc3 | 1.20 | Ethanol |
| HC49 | e_hc8 | 1.14 | Ethylene Glycol |
| PM01 | e_pm25i | 0.20 | Unspeciated primary PM2.5 - nuclei mode |
| PM01 | e_pm25j | 0.80 | Unspeciated primary PM2.5 - accumulation mode |
| PM02 | e_so4i | 0.20 | Sulfate PM2.5 - nuclei mode |
| PM02 | e_so4j | 0.80 | Sulfate PM2.5 - accumulation mode |
| PM03 | e_no3i | 0.20 | Nitrate PM2.5 - nuclei mode |
| PM03 | e_no3j | 0.80 | Nitrate PM2.5 - accumulation mode |
| PM04 | e_orgi | 0.20 | Organic Carbon PM2.5 - nuclei mode |
| PM04 | e_orgj | 0.80 | Organic Carbon PM2.5 - accumulation mode |
| PM05 | e_eci | 0.20 | Elemental Carbon PM2.5 - nuclei mode |
| PM05 | e_ecj | 0.80 | Elemental Carbon PM2.5 - accumulation mode |
| PM10-PRI | e_pm10 | 1.00 | Unspeciated Primary PM10 |

Table SM 2. Summary of model performance in capturing temperature at BAO 10m and 300m during Aug 1-15, 2014

| T (C) - 10m | OBS | PBL | | | Met IC and BC | | Initialization | | Horizontal resolution | |
|---|---|---|---|---|---|---|---|---|---|---|
| | | PBL1 | PBL2 | PBL3 | Met5 | Met6 | Init4 | Init5 | Hor5 | Hor5-12km |
| Mean | 21.67 | 22.40 | 20.95 | 21.20 | 24.06 | 23.44 | 21.59 | 24.06 | 24.06 | 24.08 |
| R | | 0.89 | 0.89 | 0.89 | 0.86 | 0.89 | 0.71 | 0.86 | 0.86 | 0.88 |
| RMSE | | 2.05 | 2.03 | 2.01 | 3.25 | 2.63 | 2.99 | 3.25 | 3.25 | 3.18 |
| MAE | | 1.56 | 1.62 | 1.59 | 2.60 | 2.05 | 2.30 | 2.60 | 2.60 | 2.53 |
| MB | | 0.74 | -0.72 | -0.46 | 2.40 | 1.77 | -0.08 | 2.40 | 2.40 | 2.41 |
| NMB | | 3.4% | -3.3% | -2.1% | 11.1% | 8.2% | -0.4% | 11.1% | 11.1% | 11.1% |
| T (C) - 300m | OBS | PBL1 | PBL2 | PBL3 | Met5 | Met6 | Init4 | Init5 | Hor5 | Hor5-12km |
| Mean | | 21.91 | 20.95 | 21.30 | 23.58 | 22.89 | 20.31 | 23.58 | 23.58 | 23.52 |
| R | | 0.76 | 0.75 | 0.72 | 0.74 | 0.78 | 0.57 | 0.74 | 0.74 | 0.75 |
| RMSE | | 2.16 | 2.14 | 2.10 | 2.79 | 2.27 | 3.09 | 2.79 | 2.79 | 2.80 |
| MAE | | 1.69 | 1.73 | 1.68 | 2.24 | 1.76 | 2.45 | 2.24 | 2.24 | 2.21 |
| MB | | 0.23 | -0.73 | -0.38 | 1.90 | 1.22 | -1.37 | 1.90 | 1.90 | 1.85 |
| NMB | | 1.1% | -3.4% | -1.8% | 8.8% | 5.6% | -6.3% | 8.8% | 8.8% | 8.5% |

Table SM 3. Summary of model performance in capturing relative humidity (RH) at BAO 10m and 300m during Aug 1-15, 2014

| RH (%)-10m | OBS | PBL | | | Met IC and BC | | Initialization | | Horizontal resolution | |
| | | PBL1 | PBL2 | PBL3 | Met5 | Met6 | Init4 | Init5 | Hor5 | Hor5-12km |
|---|---|---|---|---|---|---|---|---|---|---|
| Mean | 46.47 | 46.85 | 57.59 | 55.78 | 32.65 | 39.87 | 59.36 | 32.65 | 32.65 | 32.89 |
| R | | 0.78 | 0.69 | 0.73 | 0.63 | 0.64 | 0.53 | 0.63 | 0.63 | 0.71 |
| RMSE | | 10.89 | 16.90 | 15.13 | 19.13 | 14.95 | 22.33 | 19.13 | 19.13 | 18.15 |
| MAE | | 8.45 | 14.38 | 12.86 | 15.01 | 11.31 | 18.10 | 15.01 | 15.01 | 14.43 |
| MB | | 0.38 | 11.12 | 9.31 | -13.81 | -6.60 | 12.90 | -13.51 | -13.51 | -13.58 |
| NMB | | 0.8% | 23.9% | 20.0% | -29.7% | -14.2% | 27.7% | -29.7% | -29.7% | -29.2% |
| RH (%)-300m | OBS | PBL1 | PBL2 | PBL3 | Met5 | Met6 | Init4 | Init5 | Hor5 | Hor5-12km |
| Mean | 38.70 | 43.63 | 51.45 | 48.25 | 31.27 | 38.55 | 59.06 | 31.27 | 31.27 | 31.94 |
| R | | 0.64 | 0.59 | 0.48 | 0.53 | 0.52 | 0.41 | 0.53 | 0.53 | 0.57 |
| RMSE | | 13.06 | 17.92 | 15.25 | 12.66 | 11.14 | 28.39 | 12.66 | 12.66 | 12.11 |
| MAE | | 9.92 | 14.78 | 12.77 | 9.73 | 8.60 | 23.19 | 9.73 | 9.73 | 9.29 |
| MB | | 4.93 | 12.75 | 9.55 | -7.43 | -0.15 | 20.36 | -7.43 | -7.43 | -6.76 |
| NMB | | 12.7% | 32.9% | 24.7% | -19.2% | -0.4% | 52.6% | -19.2% | -19.2% | -17.5% |

Table SM 4 Summary of model performance in capturing wind speed and direction at BAO 10m during Aug 1-15, 2014

| | | | | PBL | | Met | | Init | | Horizontal Res. | |
|---|---|---|---|---|---|---|---|---|---|---|---|
| **Day - 10 m** | | **OBS** | **PBL1** | **PBL2** | **PBL3** | **Met5** | **Met6** | **Init4** | **Init5** | **Hor5** | **Hor5-12km** |
| Wind Speed | **Mean** | 2.46 | 2.99 | 2.68 | 2.20 | 2.63 | 2.83 | 3.30 | 2.63 | 2.63 | 2.58 |
| | **STD** | 1.25 | 1.47 | 1.55 | 1.27 | 1.41 | 1.51 | 2.02 | 1.41 | 1.41 | 1.33 |
| Wind Direction | **Mean** | 123.38 | 64.31 | 71.92 | 74.85 | 38.63 | 70.83 | 61.40 | 38.63 | 38.63 | 45.08 |
| | **STD** | 66.06 | 45.40 | 62.30 | 54.02 | 73.77 | 75.30 | 75.65 | 73.77 | 73.77 | 66.18 |
| **Night - 10 m** | | **OBS** | **PBL1** | **PBL2** | **PBL3** | **Met5** | **Met6** | **Init4** | **Init5** | **Hor5** | **Hor5-12km** |
| Wind Speed | **Mean** | 2.25 | 2.81 | 2.58 | 2.18 | 2.51 | 2.72 | 2.91 | 2.51 | 2.51 | 2.66 |
| | **STD** | 0.96 | 1.41 | 0.94 | 0.96 | 1.35 | 1.43 | 1.40 | 1.35 | 1.35 | 1.41 |
| Wind Direction | **Mean** | 222.98 | 244.07 | 243.95 | 263.07 | 226.97 | 230.93 | 160.02 | 226.97 | 226.97 | 295.43 |
| | **STD** | 50.01 | 90.68 | 69.52 | 74.66 | 83.89 | 69.81 | 87.15 | 83.89 | 83.89 | 87.30 |

Table SM 5 Summary of model performance in capturing wind speed and direction at BAO 300m during Aug 1-15, 2014

| Day - 300 m | | OBS | PBL | | | Met | | init | | Horizontal Res. | |
|---|---|---|---|---|---|---|---|---|---|---|---|
| | | OBS | PBL1 | PBL2 | PBL3 | Met5 | Met6 | Init4 | Init5 | Hor5 | Hor5-12km |
| Wind Speed | Mean | 3.23 | 3.89 | 3.51 | 2.78 | 2.88 | 3.22 | 3.83 | 2.88 | 2.88 | 2.77 |
| | STD | 2.24 | 2.15 | 2.39 | 1.61 | 1.58 | 1.81 | 2.93 | 1.58 | 1.58 | 1.47 |
| Wind Direction | Mean | 117.31 | 62.69 | 62.42 | 64.05 | 32.91 | 57.67 | 56.71 | 32.91 | 32.91 | 39.52 |
| | STD | 74.56 | 51.99 | 63.89 | 59.84 | 75.14 | 76.03 | 74.32 | 75.14 | 75.14 | 69.43 |
| Night - 300 m | | | PBL1 | PBL2 | PBL3 | Met5 | Met6 | Init4 | Init5 | Hor5 | Hor5-12km |
| Wind Speed | Mean | 3.42 | 5.00 | 4.34 | 3.80 | 4.21 | 4.60 | 5.07 | 4.21 | 4.21 | 4.89 |
| | STD | 2.59 | 2.68 | 2.95 | 2.64 | 2.64 | 2.47 | 3.07 | 2.64 | 2.64 | 3.29 |
| Wind Direction | Mean Model | 213.59 | 141.12 | 223.36 | 355.95 | 326.05 | 294.02 | 156.88 | 326.05 | 326.05 | 306.58 |
| | STD Model | 72.73 | 98.36 | 93.80 | 91.39 | 91.33 | 77.67 | 84.60 | 91.33 | 91.33 | 88.31 |

[Figure]

Figure SM 1 Average diurnal cycle of temperature, relative humidity, wind speed, and wind direction for all test sets and observation at BAO 10m. Averages are calculated for Aug 1 to 15, 2014

[Figure]

Figure SM 2 Average diurnal cycle of temperature, relative humidity, wind speed, and wind direction for all test sets and observation at BAO 100m. Averages are calculated for Aug 1 to 15, 2014

[Figure]

Figure SM 3 Average diurnal cycle of temperature, relative humidity, wind speed, and wind direction for all test sets and observation at BAO 300m. Averages are calculated for Aug 1 to 15, 2014

[Figure]

Figure SM 4. Average diurnal cycle of wind speed (WS) and direction (WD) at WC Tower and PAO sites. Averages are calculated for August 1 to 11, 2014

[Figure]

Figure SM 5 Wind speed at 10m captured by different PBL schemes. averaged from 1-August to 11-August 2014

[Figure]

Figure SM 6. Surface ethane in sim 1 (1-YFM), sim 2 (2-MjFM), sim 3 (3-MnFm) averaged from August 1 to 15, 2014

[Figure]

Figure SM 7 Differences in temperature (a), relative humidity (b), and wind speed (c) between ERA-interim and NCEP-FNL global models (Δx = X(ERA-interim) – X(NCEP-FNL)) averaged from Aug 1 to 15, 2014 using 6-hourly data

[Figure]

Figure SM 8. Sensitivity of ethane to oil and NG emission during C130-AM (a), C130-PM (b), P3-PAO AM (d), P3-PAO PM (c), P3-BAO AM (e), P3-BAO PM (f) averaged for August flights

---

## Author Response (AR2)

Reviewer 1:

*R1: The authors did a lot of effort to improve their manuscript based on my suggestions.*

*However, unfortunately, the authors did not take up the suggestion of reviewer #2.*
*I support his idea, that the analysis (one might call it evaluation) would win a lot from looking at other trace gases.*
*This is especially true, as one of the results -- at least from my point of view -- is, that we can not learn much from the comparison to ethane measurements, as the underlying ethane emissions are totally off. The aim of this manuscript is to evaluate how the transport of a trace gas (finally targeted ethane) is effected by different physical parametrisations. Or, to phrase it differently, the main question is, which physical parametrisations do lead to the most realistic transport in the boundary layer and above. But this can not be answered by looking at ethane, if the emissions are thus far off, as in the present study. Here I also agree with the second reviewer, one could have learned more from a passive tracer study.*

*All the comparisons of a) different boundary layer schemes, b) the gain of re-initialisation, c) different initial and lateral boundary data and d) the horizontal resolution would be more meaningful, if gases which sources are better constrained than ethane would be compared.*

*I think, this point, raised by the second reviewer is the major week point of the study and is the reason, why I still considered the paper "major revisions".*

Response:
We appreciate your comments. We have discussed the limitations of this work more clearly in the conclusion section:

"**We recognize that using ethane as a tracer to assess the sensitivity of WRF-Chem model to physical parametrization can be limited by the biases in the emission inventory. Conducting WRF-Chem simulations using different physical parametrization and using NMB variability can help to reduce this limitation.** The presented results reflect the challenges that one is faced with when attempting to improve emission inventories by contrasting measured with modeled concentrations, either through simple direct comparisons or more advanced methods, such as inverse modeling. Any uncertainties that arise from the model configuration will translate into the derived emission constraints, and it is important to be aware of the uncertainties resulting from different model setups. The WRF-Chem simulations and knowledge gained from this study will be used to support inverse modeling studies aimed to improve estimates of emission from oil and NG sector in the NFR."

As discussed in the response to reviewer 2, because of the multiple emission sources in the region such as transport sector, power plants, refineries, oil and NG sites, and feedlots finding a better tracer is very complicated. Using a tracer such as ethane with one dominant source, relatively long lifetime, and available rich measurement dataset enabled us to simplify the problem by focusing on one sector (i.e., oil and NG). We understand that the emission inventory underestimates the ethane emission rates but comparing the errors in different tests by using NMB variability can help us quantify the differences in the model performance caused by model parametrization. The focus of this work is to understand the

sensitivity of model errors to physical parametrization, thus we used NMB variability. This work will be followed by an inverse modeling method to constrain the ethane emission estimates based on P3 and C130 measurements. NMB variability will assist in adding confidence levels to the inverse modeling solution based on variability in model errors due to model parametrization.

R1: Some detailed points of criticism:

1) *from my point of view sect 2.3 should be Sect 2.2.1 and wiseversa, because the emission inventory belongs to the genereal setup of the WRF-Chem model, while a section "sensitivity tests" should provide information on the modifications of the general setup.*

Response:

We think that discussing different sensitivity tests (section 2.2.1.) right after describing general model set-ups and configurations can help the readers to better recognize the modifications in the model configurations in each sensitivity test. Having emission inventory in another subsection can help readers to access this information easier.

2) *p.5, l. 15/16: "WRF-Chem provides a processed version ...." YOu honestly do not mean, that the model itself provides Emission data? Most probably the work is done by some institution ...?*

Response:

we changed the sentence to:

**"A processed version of NEI-2011 is available to the users ...."**

3) *Table 2: Provide a better description of the table. Necessary information: sensitivity tests divided by horizontal lines, The same simulation is mentions multiple times ....*
*Additionally, if I understand correctly Em7 is the same as Met5 / Init5 etc. Therefore it is rather confusing to add "+ Megan" here. This sounds as if Megan was only used in the to Em simulations.*

Response:

We changed the caption to:

**"Table 2. Summary of WRF-Chem configurations for sensitivity tests designed for this study. Sensitivity tests are divided by horizontal lines"**

Some of the simulations were used in multiple tests thus was mentioned in table 2 multiple times. Megan emissions was only used in Em7 and Em8 simulations.

4) *Profile comparisons for ethane:*
*4a) provide somewhere the information of the definition of the "vertical bins" (0-1.5 km, 1.5-2 km ... ???)*

Response:

We added details to the figure 4 and 5 captions:

**"Measurement pointes were binned based on their elevation above the ground in 500m intervals. The first bin includes all measurements below 1.5km and the last bin includes all measurements above 3km"**

*4b) as the Ethane emissions are much too low, for the comparison of the shape of the profiles (which provide the information about the vertical transport), it would be better to compare "relative profiles", i.e. normed to their respective median, surface (or what else) concentration.*

Response:

We created figures using relative values normalized to the measured ethane concentration median at the surface (Figure 1). However, we believe that using the absolute values of ethane concentration at different altitudes better represents the vertical distribution of ethane concentration and better highlights the mixing height.

[Figure]

*Figure 1 vertical ethane concentration distribution normalized to the measured surface ethane concentration at the surface*

*5) throughout the paper: "boundary conditions" is a very general term. This could be emissions, a land-cover data set or "lateral boundary conditions" therefore add the word "lateral" when talking about "lateral boundary conditions".*

Response:

We added the term "lateral" to "boundary conditions"

*6) Fig. 3 lacks panels j,k,l and a colourbar.*

Response:

We made the added the color bar and modified the caption

[revised manuscript text omitted]